# Discretization Invariant Networks for Learning Maps between Neural Fields

**Clinton Wang**  *clintonw@csail.mit.edu*
*MIT CSAIL*

**Polina Golland**  *polina@csail.mit.edu*
*MIT CSAIL*

**Reviewed on OpenReview:** *https://openreview.net/forum?id=CpYBAqDgmz*

## Abstract

With the emergence of powerful representations of continuous data in the form of neural fields, there is a need for discretization invariant learning – an approach for learning maps between functions on continuous domains without being sensitive to how the function is sampled. We present a new framework for understanding and designing discretization invariant neural networks (DI-Nets), which generalizes many discrete networks such as convolutional neural networks as well as continuous networks such as neural operators. Our analysis establishes upper bounds on the deviation in model outputs under different finite discretizations, and highlights the central role of point set discrepancy in characterizing such bounds. This insight leads to the design of a family of neural networks driven by numerical integration via quasi-Monte Carlo sampling with discretizations of low discrepancy. We prove by construction that DI-Nets universally approximate a large class of maps between integrable function spaces, and show that discretization invariance also describes backpropagation through such models. Applied to neural fields, convolutional DI-Nets can learn to classify and segment visual data under various discretizations, and sometimes generalize to new types of discretizations at test time. Code: `https://github.com/clintonjwang/DI-net`.

## 1 Introduction

Neural fields (NFs), which encode signals as the parameters of a neural network, have many useful properties. NFs can efficiently store and stream continuous data (Sitzmann et al., 2020b; Takikawa et al., 2022; Cho et al., 2022), represent and render detailed 3D scenes at lightning speeds (Müller et al., 2022; Barron et al., 2023; Kerbl et al., 2023), and integrate data from a wide range of modalities (Gao et al., 2021; 2022). NFs are thus an appealing data representation for many applications, but learning downstream tasks such as NF classification, segmentation, or generation remains a challenging problem. Their continuous domain makes NFs unsuitable as inputs to traditional neural network architectures that are designed for discrete pixel or voxel grids.

Current approaches for training networks on a dataset of NFs predominantly focus on learning in NF parameter space (Tancik et al., 2020a; Dupont et al., 2022; Mehta et al., 2021), but such approaches have two major disadvantages: 1) once trained on the parameter space of one type of NF, they become incompatible with other types; 2) they are unsuitable for important classes of NFs whose parameters extend beyond a neural network, such as those with voxel (Sun et al., 2021; Alex Yu and Sara Fridovich-Keil et al., 2021), octree (Yu et al., 2021), hash table (Müller et al., 2022; Takikawa et al., 2022; Barron et al., 2023) or other (Kerbl et al., 2023) components. We instead view NFs as black box vector-valued functions, and hence performing inference on NFs can be understood as learning operators on a function space. Given that an algorithm can only evaluate an NF at a finite number of points, we ask how a learning algorithm should sample each NF on

its continuous domain, and how to design a model whose output is largely independent of how the sample points are chosen, a property called *discretization invariance.*

Current analyses of discretization invariance are limited to showing the existence of operators that converge in the limit of discretizations with infinite points (Kovachki et al., 2021b), or they demand that the function spaces be constrained to those that can be discretized losslessly (Bartolucci et al., 2023). They do not examine the effect of the discretization itself on the approximation error in the general case. Characterizing this effect is particularly relevant in the context of neural fields, which permit many different types of discretizations and are often queried repeatedly under slightly different discretizations in applications such as novel view synthesis. In this paper we describe discretization invariant neural networks (DI-Nets), a broad class of networks for learning maps between integrable function spaces such as those represented by neural fields. We explore how different discretizations yield different behaviors in the finite case, and establish the central role of point set discrepancy – points in a limiting discretization must be evenly distributed to achieve convergence. By specifying layers as integrals over parametric functions of the input field, DI-Nets have access to powerful numerical integration techniques such as quasi-Monte Carlo sampling, which yields fast convergence by choosing low discrepancy discretizations. DI-Nets encompass continuous networks such as neural operators (Kovachki et al., 2021b), while also extending discrete networks that act on pixels, point clouds, and meshes.

**Summary of contributions:** we analyze discretization invariance in the finite sample case, where discrepancy and equidistributed sequences play a central role. Our analysis gives rise to a large family of discretization invariant networks (DI-Nets) that universally approximate a large class of maps between function spaces. We show that backpropagation through DI-Nets also yields discretization invariant gradients. We probe the limits of discretization invariant networks in practice, demonstrating that convolutional DI-Nets can learn classification and dense prediction tasks on neural fields under a range of different discretizations. We show that DI-Net has some ability to generalize to new discretizations at test time, whereas maps learned by pre-trained discrete networks collapse under slight perturbations of the discretization.

## 2 Related Work

**Neural fields**  Neural fields (also called implicit neural representations) are neural networks that can be trained to capture a wide range of continuous data with high fidelity. They are usually parameterized as MLPs, sometimes with additional components such as features stored in voxel (Sun et al., 2021), octree (Yu et al., 2021) or hash table (Müller et al., 2022; Takikawa et al., 2022; Barron et al., 2023) structures. Alternatively, neural fields can be represented directly as a set of parameters optimized directly with gradient descent (Alex Yu and Sara Fridovich-Keil et al., 2021; Kerbl et al., 2023). The most prominent domains include shapes (Park et al., 2019; Mescheder et al., 2018), objects (Niemeyer et al., 2020; Müller et al., 2022), and 3D scenes (Mildenhall et al., 2020; Sitzmann et al., 2021), but previous works also apply NFs to gigapixel images (Martel et al., 2021), volumetric medical images (Corona-Figueroa et al., 2022), acoustic data (Sitzmann et al., 2020b; Gao et al., 2021), tactile data (Gao et al., 2022), depth and segmentation maps (Kundu et al., 2022), and 3D motion (Niemeyer et al., 2019).

**Learning on neural fields**  Hypernetworks and modulation networks were developed for learning on neural fields, and have been demonstrated on tasks including generative modeling, data imputation, novel view synthesis and classification (Sitzmann et al., 2020b; 2021; Tancik et al., 2020a; Sitzmann et al., 2019; 2020a; Mehta et al., 2021; Chan et al., 2021; Dupont et al., 2021; 2022). Hypernetworks use meta-learning to learn to produce the MLP weights of desired output NFs, while modulation networks predict modulations that can be used to transform the parameters of an existing NF or generate a new NF. An alternative approach uses the derivative networks of the NF (Xu et al., 2022), training an MLP that learns a mapping between first through $k$th order derivatives of the NF and the desired output signal. Both of these approaches require that the input NFs are entirely MLPs, and do not generalize to new architectures once trained. An alternative approach for learning 3D NF→NF maps evaluates an input NF at fixed grid points, produces features at the same points via a U-Net, raytraces interpolated features, then uses an MLP decoder to produce output values from arbitrary camera views (Vora et al., 2021).

**Group invariant neural networks**   There is a rich literature exploring the design and training of neural networks for learning maps between function spaces subject to symmetries such as permutation invariance (Zweig & Bruna, 2021; Zaheer et al., 2017), rotational invariance (Cheng et al., 2018), or more general group invariances (Yarotsky, 2022; Lyle et al., 2020). Discretization invariance must be treated differently from group invariance as discretizations lack all the properties of groups (associativity, identity element, and inverse element). Moreover, producing identical outputs under arbitrary discretizations is impossible on all but the most trivial function spaces, hence discretization invariance must be described in an approximate or limiting sense. Since group actions map between different discretizations, discretization invariance can be seen as a weaker yet more general form of group invariance. Discretization invariance may be more useful in settings where the task of interest should be able to be solved under a wide range of discretizations which are not related by a single group.

**Approximation capabilities of neural networks**   A fundamental result in approximation theory is that the set of single-layer neural networks is dense in a large space of functionals including $L^p(\mathbb{R}^n)$ (Hornik, 1991). Subsequent works designed constructive examples using various non-linear activations (Chen et al., 1995; Chen & Chen, 1993). While this result is readily extended to multi-dimensional outputs, existing approximation results for the case of infinite dimensional outputs (e.g., $L^p(\mathbb{R}^n) \to L^p(\mathbb{R}^n)$) do not explicitly characterize the contribution of data discretization to the approximation error (Bhattacharya et al., 2020; Lanthaler et al., 2022; Kovachki et al., 2021b;a). Universal approximation results for operator learning frameworks typically quantify the approximation error in terms of the class of functions being approximated or the type of layers used in the network (Kovachki et al., 2021b; Bartolucci et al., 2023; Kissas et al., 2022; Prasthofer et al., 2022) rather than the choice of discretization, a gap that we seek to fill in this work.

**Discretization invariant networks**   Networks that are agnostic to the discretization of the data domain has been explored primarily in the context of learning operators between function spaces. Hilbert space PCA, DeepONets and neural operators are tailored to solve partial differential equations efficiently in a manner that converges as more sensors are added and the discretization of the input space is refined (Bhattacharya et al., 2020; Lu et al., 2021a; Li et al., 2020a; Kovachki et al., 2021b). In the context of learning on surface meshes, DiffusionNet (Sharp et al., 2022) also defines discretization invariance as convergent behavior in the limit of infinite sample points (mesh refinement). The recent operator learning framework ReNO (Bartolucci et al., 2023) starts with the assumption that there exists a lossless discretization of the input and output function spaces which is known a priori (e.g., they are bandlimited functions), then establishes necessary conditions for learning (lossless) operators between such spaces. Other works are concerned with more practical aspects of operator learning: LOCA (Kissas et al., 2022) leverages attention to more efficiently learn correlations between related points in output space; NOMAD (Seidman et al., 2022) aims to increase expressivity given finite basis elements by equipping neural operators with nonlinear decoders; VIDON (Prasthofer et al., 2022) builds on DeepONet to accommodate arbitrary locations and numbers of input and output query points (as does our framework); other extensions of DeepONet and neural operators refine the original works to improve learning efficiency or generalizability on certain domains (Li et al., 2020b; Lu et al., 2021b; Lee et al., 2022).

**Continuous convolutions**   At the core of many discretization invariant approaches is the continuous convolution, which also provides permutation invariance, translation invariance and locality. Its applications include modeling point clouds (Wang et al., 2021; Boulch, 2019), graphs (Fey et al., 2017), fluids (Ummenhofer et al., 2019), and sequential data (Romero et al., 2021b;a), where there is typically no choice of how the data should be discretized. This work focuses on the effect of different discretizations, proposes quasi-Monte Carlo as a canonical method of generating discretizations, and can produce neural fields as output.

## 3   Discretization Invariant Learning

### 3.1   Discretization Invariance

Let $\Omega$ be a bounded measurable subset of a $d$-dimensional compact metric space, for example a compact subset of $\mathbb{R}^d$ or a $d$-dimensional manifold. Consider the space of vector-valued functions of bounded variation $\mathcal{F}_c = \{f : \Omega \to \mathbb{R}^c : \int_\Omega \|f\|^2 d\mu < \infty \text{ and } V(f) < \infty\}$, where the variation $V(f)$ measures how much the

function fluctuates over its domain. The variation of a 1D function $f \in C^1([a, b])$ is given by:

$$V(f) = \int_a^b |f'(x)| dx, \tag{1}$$

and more general definitions are given in Appendix A.1.

The *discretization* of a function is a finite point set $X \subset \Omega$ on which it is queried. We say that a map $\mathcal{H} : \mathcal{F}_c \to \mathbb{R}^n$ is discretizable if it induces a discrete operator $\hat{\mathcal{H}}^X : \mathcal{F}_c \to \mathbb{R}^n$, which seeks to replicate the behavior of the original map but depends only on the input's values at $X$. Ideally we would be able to design maps that are truly invariant to the choice of discretization, meaning that all its discrete operators are identical. But such idealized discretization invariance is only possible on the most trivial function spaces, and thus a more practical definition of discretization invariance is necessary:

**Definition 3.1.** A discretizable map $\mathcal{H} : \mathcal{F}_c \to \mathbb{R}^n$ is discretization invariant if there exists constant $k > 0$ such that for every discretization $X$ and function $f$, $\left\| \mathcal{H}[f] - \hat{\mathcal{H}}^X[f] \right\|_1 \leq kV(f)D(X)$ where $D(X)$ is the discrepancy of $X$. A map $\bar{\mathcal{H}} : \mathcal{F}_c \to \mathcal{F}_n$ is discretization invariant if $\bar{\mathcal{H}}[\cdot](x)$ is uniformly discretization invariant for all $x \in \Omega$.

This definition establishes an upper bound on the deviation between any two discretizations by a simple application of the triangle inequality. The discrepancy of a discretization is lower for dense, evenly distributed points. For a 1D point set on domain $\Omega = [a, b]$, it is given by:

$$D(\{x_i\}_{i=1}^N) = \sup_{a \leq c \leq d \leq b} \left| \frac{|\{x_1, \ldots, x_N\} \cap [c, d]|}{N} - \frac{d - c}{b - a} \right|. \tag{2}$$

See Appendix A.1 for general definitions of discrepancy. The product of variation and discrepancy is precisely the upper bound in the celebrated Koksma–Hlawka inequality, which bounds the difference between the integral of a function $h \in L^2(\Omega)$ and its sample mean on any point set $X \subset \Omega$:

$$\left| \frac{1}{|X|} \sum_{x' \in X} h(x') - \int_\Omega h(x)\, dx \right| \leq V(h)\, D(X). \tag{3}$$

### 3.2 Discretization Invariant Layers

This naturally leads to a family of **discretization invariant (DI) layers** specified as the integral of a parametric map over an input function:

$$\mathcal{H}_\phi : f \mapsto \int_\Omega H_\phi[f](x) dx, \tag{4}$$

whose discrete operator is simply its sample mean:

$$\hat{\mathcal{H}}_\phi^X : f \mapsto \frac{1}{|X|} \sum_{x \in X} H_\phi[f](x). \tag{5}$$

The parametric map $H_\phi[f]$ can have two forms:

- In vector-valued DI layers, $\hat{\mathcal{H}}_\phi^X : \mathcal{F}_c \to \mathbb{R}^n$ and $H_\phi[f](x) = h_\phi(x, f(x)) \in \mathbb{R}^n$. Such layers include global pooling and learned inner products, and could be used as one of the final layers in a classification network.

- In function-valued DI layers, $\hat{\mathcal{H}}_\phi^X : \mathcal{F}_c \to \mathcal{F}_n$ and $H_\phi[f](x) : x' \mapsto h_\phi(x, x', f(x), f(x')) \in \mathbb{R}^n$ for all $x' \in \Omega$. Such layers include continuous convolutions, deconvolutions, and self-attention layers.

In each case, $h_\phi$ must be bounded and continuous in all its variables so that outputs remain of bounded variation (hence satisfying our definition of discretization invariance). $h_\phi$ must also be differentiable w.r.t. $\phi$ to enable backpropagation, and Gateaux differentiable w.r.t. $f$ to make its gradients discretization invariant, as we discuss in Section 3.3. We consider even more general forms of DI layers in Appendix A.2.

The upper bound in Eq. (3) points to two levers for reducing the approximation error of a given layer. We can design $H_\phi$ to have low variation for most $f$, for example by imposing Lipschitz regularization, but there is a tradeoff between reducing variation and maintaining the layer's ability to capture information relevant to the downstream task. We can also choose $X$ to have low discrepancy by using fast methods for generating low discrepancy sequences. Obtaining the sample mean in this way is called quasi-Monte Carlo (QMC), a numerical integration method with favorable convergence rates compared to standard Monte Carlo (Caflisch, 1998). In this work, we assume that the discretization is chosen once without *a priori* knowledge of the function, making QMC optimal. But we note that tighter bounds on the approximation error can be attained by choosing a different discrete operator based on quadrature, and/or by refining the discretization after initial evaluations of the operator. Replacing $1/|X|$ in the sample mean with quadrature weights will achieve better convergence when the discrepancy of $X$ is high, and adaptive quadrature updates the discretization to attain specific error bounds at inference time, which can be valuable in applications requiring robustness or verification. Additionally rejection sampling can be used for the Monte Carlo method under non-uniform measures.

A **discretization invariant network** (DI-Net) is a directed acyclic graph of DI layers as well as pointwise layers.[1] A pointwise layer is a bounded continuous scalar function applied to each point in $\Omega$, and includes nonlinear activations, batch normalization, as well as addition or concatenation of NFs. Since pointwise layers preserve an NF's property of bounded variation, DI-Net is discretization invariant. A prototypical DI-Net for classification might consist of NF-valued DI layers separated by normalization and activation layers, and end with a vector-valued DI layer followed by softmax.

### 3.3 Convergence under Equidistributed Discretizations

From our definition of discretization invariance, it is clear that the approximation error of DI layers converges to 0 under sequences of discretizations whose discrepancy tends to 0. We call such a sequence of discretizations an *equidistributed* discretization sequence. A simple way to generate an equidistributed discretization sequence is to truncate any equidistributed sequence of points to the first $N$ terms for each $N \in \mathbb{N}$. Quasi-Monte Carlo (QMC) sampling efficiently generates equidistributed sequences on a wide range of domains.

We can also ask whether the DI-Net's learning algorithm is also discretization invariant: specifically whether its discretized gradients are convergent, and to what value they converge. Consider a vector-valued DI layer $\mathcal{H}_\phi$. The derivatives of its output w.r.t. its weights $\phi = (\phi_1, \ldots, \phi_K)$ can be shown to converge under any equidistributed discretization sequence $\{X_N\}_{N \in \mathbb{N}}$. Denote $\phi + \tau e_k = (\phi_1, \ldots, \phi_{k-1}, \phi_k + \tau, \phi_{k+1}, \ldots, \phi_K)$.

$$\lim_{N \to \infty} \frac{\partial}{\partial \phi_k} \hat{\mathcal{H}}_\phi^N[f] = \lim_{N \to \infty} \frac{\partial}{\partial \phi_k} \left( \frac{1}{|X_N|} \sum_{x \in X_N} H_\phi[f](x) \right) \tag{6}$$

$$= \lim_{N \to \infty} \lim_{\tau \to 0} \left( \frac{1}{\tau N} \sum_{j=1}^{N} H_{\phi + \tau e_k}[f](x) - H_\phi[f](x) \right) \tag{7}$$

$$= \lim_{\tau \to 0} \frac{1}{\tau} \int_\Omega H_{\phi + \tau e_k}[f](x) - H_\phi[f](x) \, dx \tag{8}$$

$$= \lim_{\tau \to 0} \frac{\mathcal{H}_{\phi + \tau e_k}[f] - \mathcal{H}_\phi[f]}{\tau} \tag{9}$$

$$= \frac{\partial}{\partial \phi_k} \mathcal{H}_\phi[f], \tag{10}$$

---

[1]As its first layer a DI-Net that maps vectors to functions may require a vector decoder: a map $\mathbb{R}^n \to \mathcal{F}_c$ specified by $n$ elements of a basis on $\mathcal{F}_c$ or interpolation of $n/c$ points (see Appendix C for examples).

where (8) uses the Moore-Osgood theorem. The case of a function-valued DI layer proceeds identically, showing this condition holds for the derivatives at each point on the output domain. Thus the discretized gradient converges to the Jacobian of $\mathcal{H}_\phi$ w.r.t. each parameter, which is finite by differentiability and boundedness of $H_\phi$.

Describing the derivative of the layer's output w.r.t. the input function is more nuanced, since pointwise derivatives are not sufficient to represent backpropagation in the continuous case. We must instead use the Gateaux derivative, which describes the linear change in a map between functions given an infinitesimal change in the input function. We prove the following in Appendix B.1:

**Lemma 3.2.** *For every $f \in \mathcal{F}_1$ and fixed $\tilde{x} \in \Omega$, we can design a sequence of bump functions $\{\psi_{\tilde{x}}^N\}_{N \in \mathbb{N}}$ which is 1 in a small neighborhood around $\tilde{x}$ and vanishes at each $X_N \backslash \{\tilde{x}\}$, such that:*

$$\lim_{N \to \infty} \frac{\partial}{\partial f(\tilde{x})} \hat{\mathcal{H}}_\phi^{X_N}[f] = \lim_{N \to \infty} d\mathcal{H}_\phi[f; \psi_{\tilde{x}}^N], \tag{11}$$

*where $d\mathcal{H}_\phi[f; \psi_{\tilde{x}}^N] = \lim_{t \to 0} \frac{\mathcal{H}_\phi[f + t\psi_{\tilde{x}}^N] - \mathcal{H}_\phi[f]}{t}$ is the Gateaux derivative when $f$ is perturbed in the direction of $\psi_{\tilde{x}}^N$.*

Having established convergence of the discretized gradients w.r.t. both parameters and inputs, we then note that many common loss functions and regularizers generalize naturally to the continuous domain as bounded continuous maps $\mathcal{F}_c \to \mathbb{R}$ (e.g., L2 regularization) or $\mathcal{F}_c \times \mathcal{F}_c \to \mathbb{R}$ (e.g., mean squared error), so discretization invariance readily extends to gradients with respect to loss terms. Finally, using the chain rule for Gateaux derivatives, we can then show that all discretized gradients obtained during backpropagation through a DI-Net are convergent. These results lead to the following statement (detailed proof in Appendix B.2):

**Theorem 3.3.** *A DI-Net permits backpropagation with respect to its input and all its learnable parameters. The discretized gradients converge under any equidistributed discretization sequence.*

## 4 Universality of DI-Nets

We observed that the approximation error can be made arbitrarily small by choosing a discretization with sufficiently small discrepancy. Functions of bounded variation are piecewise smooth, hence they can be represented as the integral of some function. Our parameterization of DI layers as integrals DI-Nets are universal approximators in the following sense:

**Theorem 4.1.** *For every Lipschitz continuous map $\mathcal{R} : \mathcal{F}_c \to \mathcal{F}_n$, $c, n \in \mathbb{N}$, there exists a DI-Net that approximates it to arbitrary accuracy w.r.t. a finite measure $\nu$ on $\mathcal{F}_c$. As a corollary, every Lipschitz continuous map $\mathcal{F}_c \to \mathbb{R}^n$ or $\mathbb{R}^n \to \mathcal{F}_c$ can also be approximated by some DI-Net.*

Appendix B.3 provides a full proof. Here we provide a high-level sketch of the $\mathcal{F}_1 \to \mathcal{F}_1$ case, where there exists a DI-Net $\mathcal{T}$ that satisfies:

$$\int_{\mathcal{F}_1} \|\mathcal{R}(f) - \mathcal{T}(f)\|_{L^1(\Omega)} \nu(df) < \epsilon. \tag{12}$$

1. Fix a discretization $X$ of sufficiently low discrepancy to approximate any function in $\mathcal{F}_1$ to desired accuracy. Let $N = |X|$, the number of points in the discretization.

2. Let $\pi$ be a projection operator that maps every function $f \in \mathcal{F}_1$ to $\mathbb{R}^N$ by selecting its $N$ values along the discretization. Through $\pi$, the measure $\nu$ on $\mathcal{F}_1$ induces a measure $\mu$ on $\mathbb{R}^N$.

3. Any function in $L^2(\mathbb{R}^N)$ can be approximated by covering the volume under the graph of the function with almost disjoint rectangles, and then at inference time summing the heights of the rectangles at the given $\mathbb{R}^N$ input. A multilayer perceptron (MLP) can then approximate this rectangle cover to arbitrary accuracy with sufficiently steep slopes at their boundaries (Lu et al., 2017).

4. The map $J : \pi f \mapsto \mathcal{R}[f](x)$ is in $L^2(\mathbb{R}^N)$ for each $x \in X$. We can mimic the MLP from step 3 with a DI-Net that specifies the desired connections on $\Omega$ using element-wise products with cutoff functions and

linear combinations of channels. The cutoff functions extract the input values along $X$ into separate channels, and the weights of the channels match the weights of the hypothetical MLP, projecting the result to the appropriate $x$. We present the design of such a network in Algorithm 1, with additional details in the Appendix.

5. We repeat this construction $N$ times to specify values at each of the $N$ output points in $X$, and map all other output points to the value of the closest specified point. Then we have fully specified the desired behavior of $f \mapsto \mathcal{R}[f]$ to desired accuracy w.r.t. the measure $\nu$.

---

**Algorithm 1:** DI-Net approximation of $J : \pi f \mapsto \mathcal{R}[f](x)$

---

**Input:** target function $J$, discretized input $\mathbf{f} = \{(\pi f)_k\}_{k=1}^n$, $L_1$ tolerance $\epsilon/2$

Choose rectangles $R_i^+ = [a_{i1}^+, b_{i1}^+] \times \cdots \times [a_{in}^+, b_{in}^+] \times [\zeta_i^+, \zeta_i^+ + y_i^+]$ covering the graph of
$J^+(\mathbf{f}) \triangleq \max(0, J(\mathbf{f}))$ and rectangles $R^-$ covering the graph of $J^-(\mathbf{f}) \triangleq \max(0, -J(\mathbf{f}))$ (precise conditions stated in (82));

$\delta^+ \leftarrow \frac{1}{2}\left(1 - (1 - \frac{\epsilon}{8}\left(\|J^+\|_1 + \frac{\epsilon}{16}\right)^{-1})^{1/n}\right)$;

$\delta^- \leftarrow \frac{1}{2}\left(1 - (1 - \frac{\epsilon}{8}\left(\|J^-\|_1 + \frac{\epsilon}{16}\right)^{-1})^{1/n}\right)$;

$x \leftarrow (0, 0, 1, 0, 0)$;

**for** rectangle $R_i^+ \in R^+$ **do**

    **for** dimension $k \in 1 : n$ **do**

        $x \leftarrow (\mathbf{f}_k - b_{ik}^+ + \delta(b_{ik}^+ - a_{ik}^+), \mathbf{f}_k - a_{ik}^+, x_3, x_4, x_5)$;

        $x \leftarrow \texttt{ReLU}(x)$;

        $x \leftarrow (\delta - x_1, \delta - x_2, x_3, x_4, x_5)$;

        $x \leftarrow \texttt{ReLU}(x)$;

        $x \leftarrow (0, 0, x_3(x_1 - x_2)/\delta, x_4, x_5)$;

    **end**

    $x \leftarrow (0, 0, 1, y_i^+ x_3 + x_4, x_5)$;

**end**

**for** rectangle $R_i^- \in R^-$ **do**

    **for** dimension $k \in 1 : n$ **do**

        $x \leftarrow (\mathbf{f}_k - b_{ik}^- + \delta(b_{ik}^- - a_{ik}^-), \mathbf{f}_k - a_{ik}^-, x_3, x_4, x_5)$;

        $\ldots$;

    **end**

    $x \leftarrow (0, 0, 1, x_4, y_i^- x_3 + x_5)$;

**end**

**Output:** $x_4 - x_5$

---

This construction suggests that the complexity of representing a given operator is linked to the minimum number of rectangles necessary to cover its pointwise graph ($J$ in step 4) with desired tolerance, and the degree to which this graph is similar for neighboring points in the output domain. Intuitively, this means that it is easier to represent operators in which each point in the output is influenced by relatively few points in the input domain, and where there is shared structure in this dependency. Thus DI-Net design can be guided by strategies for imposing structure on how different points influence each other: for example, convolutions produce translation invariant outputs that depend on the input function locally; attention layers produce outputs using sparse disconnected regions of the input function; Fourier neural operators (Li et al., 2020a) produce outputs driven by low frequency patterns in the input function.

## 5 Design and Implementation of DI-Nets

DI-Nets encompass a very large family of neural networks: we only specify the architecture as a directed acyclic graph, and DI layers include a wide variety of network layers. We first discuss their application to

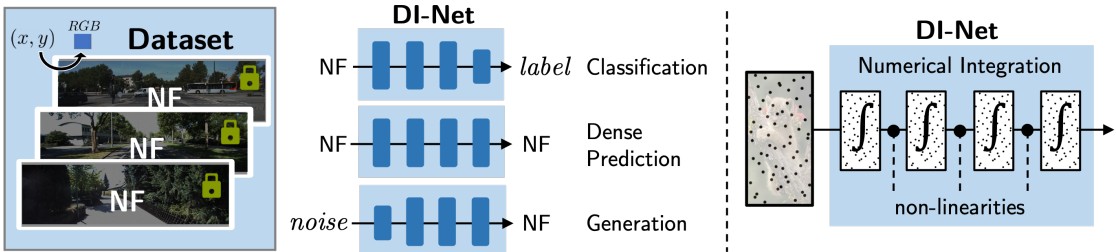

Figure 1: A discretization invariant network (DI-Net) treats neural fields as vector-valued functions. It evaluates an input field on a point set (discretization) which is used to perform numerical integration throughout the network. DI-Nets are interoperable between all types of NFs and can be trained on a wide range of tasks.

neural fields, which can be treated as function spaces to achieve parameterization-agnostic learning. We then discuss the connection between DI-Nets and neural operators (Kovachki et al., 2021b; Lu et al., 2021a), which can learn general maps between function spaces but in practice are designed to solve partial differential equations. Next we show that DI-Nets also encompass continuous adaptations of networks designed on discrete domains such as convolutional neural networks (CNNs). In the same way that neural fields extend signals on point clouds, meshes, grids, and graphs to a compact metric space, DI-Nets extend networks that operate on discrete signals by converting every layer to an equivalent discretizable map. Lastly we describe training and inference pipelines for DI-Nets.

## 5.1 Learning on Neural Fields

The parameterization of neural fields in terms of multi-layer perceptrons guarantees that they produce integrable functions of bounded variation over a compact domain, which is the function space we considered throughout Section 3. Specifically, a $d$-dimensional neural field with $c$ channels represents a vector-valued function from a $d$-dimensional domain $\Omega$ to $\mathbb{R}^c$. An occupancy network (Mescheder et al., 2018) is 3-dimensional and has 1 channel. NeRF (Mildenhall et al., 2020) is 5-dimensional (3 world coordinates and 2 view angles) and has 4 channels. We assume that neural field evaluation yields pointwise values, i.e. the point spread function of the underlying signal is a delta function. It is possible to accommodate non-trivial point spread functions, but this is beyond the scope of this work.

By treating neural fields as functions, this deep learning framework is parameterization-agnostic – the behavior of DI layers does not depend on the parameterization of the NFs that it takes as input. This property allows DI-Nets to be applied to a mixture of neural field types, which is not possible with most approaches for learning on neural fields such as hypernetworks or modulation-based networks.

The action of a function-valued DI layer produces a neural field with parameters $(\theta, \phi)$, and retaining all the DI-Net parameters can lead to an output NF with greatly inflated parameters, making it inefficient to evaluate repeatedly. This is unsuitable for applications requiring the output to be sampled several times at different discretizations, or where the output NF needs to be stored in its entirety. To solve this problem we can reparameterize the output NF by storing only the last few layers of the DI-Net alongside the discretized input activations, and adapt the discretizations as needed in these last few layers only. This approach is reminiscent of strategies that use discretized outputs of a (non-DI) neural network as parameters of an output neural field Vora et al. (2021).

## 5.2 Connection to Neural Operators

Neural Operators (Kovachki et al., 2021b) are defined in terms of a pointwise lifting operator $P : \mathbb{R}^{c_{in}} \to \mathbb{R}^{d_0}$, a pointwise projection operator $Q : \mathbb{R}^{d_T} \to \mathbb{R}^{c_{out}}$, and kernel operators $\{K_t\}_{t=1}^T$, $K_t : L^2(\Omega; \mathbb{R}^{d_{t-1}}) \to L^2(\Omega; \mathbb{R}^{d_t})$ associated with non-linear activations $\sigma_t : \mathbb{R}^{d_{t-1}} \to \mathbb{R}^{d_t}$, weight matrices $W_t \in \mathbb{R}^{d_{t-1}} \times \mathbb{R}^{d_t}$ and biases

$b_t : \Omega \to \mathbb{R}^{d_t}$. Then the network becomes $G : \mathbb{R}^{c_{in}} \to \mathbb{R}^{c_{out}}$:

$$G = Q \circ \sigma_T(W_T + K_T + b_T) \circ \cdots \circ \sigma_1(W_1 + K_1 + b_1) \circ P, \tag{13}$$

Because these layers are all DI layers or pointwise layers, neural operators represent an instantiation of DI-Nets, and are an effective choice if the task involves solving partial differential equations. Alternative designs become necessary when the input and output domains do not match or when the task can benefit from multi-scale architectures.

## 5.3 Convolutional DI-Nets

We describe how to design DI-Nets that generalize the behavior of convolutional neural networks to the continuous domain. The resulting convolutional DI-Net can be initialized directly with the weights of a pre-trained CNN, as we investigate in Section 6.4. We note that there is a rich literature of methods for designing improved continuous convolutions (Qi et al., 2017; Wang et al., 2021; Boulch, 2019) which would be helpful for scaling convolutional DI-Nets, although we do not use these methods so that the connection between DI-Nets and discrete CNNs is more direct.

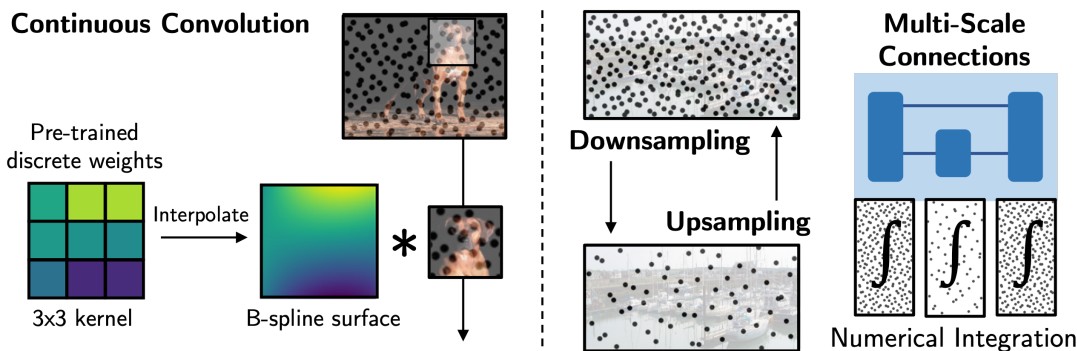

Figure 2: Convolutional DI-Nets generalize convolutional neural networks to arbitrary discretizations of the domain. Low discrepancy point sets used in quasi-Monte Carlo integration are amenable to the multi-scale structures often found in discrete networks. Convolutional DI-Nets may be initialized directly from pre-trained CNNs.

**Convolution layer** For a measurable $S \subset \Omega$ and a polynomial basis $\{p_j\}_{j \geq 0}$ that spans $L^2(S)$, $S$ is the support of a polynomial convolutional kernel $K_\phi : \Omega \times \Omega \to \mathbb{R}$ defined by:

$$K_\phi(\boldsymbol{x}, \boldsymbol{x}') = \begin{cases} \sum_{j=1}^n \phi_j p_j(\boldsymbol{x} - \boldsymbol{x}')^j & \text{if } \boldsymbol{x} - \boldsymbol{x}' \in S \\ 0 & \text{otherwise.} \end{cases} \tag{14}$$

for some chosen $n \in \mathbb{N}$. A convolution is the linear map $\mathcal{H}_\phi : \mathcal{F}_1 \to \mathcal{F}_1$ given by:

$$\mathcal{H}_\phi[f] = \int_\Omega K_\phi(\cdot, \boldsymbol{x}') f(\boldsymbol{x}') d\boldsymbol{x}'. \tag{15}$$

If $c_{\text{in}}$ is the number of input channels and $c_{\text{out}}$ the number of output channels, then the convolution layer can aggregate information across channels using $c_{\text{in}} c_{\text{out}}$ different filters.

An MLP convolution is defined similarly except the kernel becomes $\tilde{K}_\phi(\boldsymbol{x}, \boldsymbol{x}') = \text{MLP}(\boldsymbol{x} - \boldsymbol{x}'; \phi)$ in the non-zero case. While MLP kernels are favored over polynomial kernels in many applications due to their expressive power (Wang et al., 2021), polynomial bases can be used to construct filters satisfying desired properties such as group equivariance (Cohen & Welling, 2016a;b), $k$-Lipschitz continuity, or boundary conditions.

The input and output discretizations of the layer can be chosen independently, allowing for padding or striding. The input discretization fully determines which points on $S$ are evaluated. To transfer weights from a discrete convolutional layer, $K$ can be parameterized as a rectangular B-spline surface that interpolates the weights (Fig. 2 left). We use a 2nd order B-spline for $3 \times 3$ filters and 3rd order for larger filters. We use deBoor's algorithm to evaluate the spline at intermediate points.

**Pointwise channel mixing**  Linear combinations of channels can be used similarly to $1 \times 1$ convolutional layers in discrete networks. For learned scalar weights $W_{ij}$ and biases $b_j$ and all $x \in \Omega$:

$$g_j(x) = \sum_{i=1}^{c_{\text{in}}} W_{ij} f_i(x) + b_j. \tag{16}$$

**Normalization**  All forms of layer normalization readily generalize to the continuous setting by estimating the statistics of each channel with numerical integration, then applying point-wise operations. These layers typically rescale each channel to have some learned or fixed mean $m_i$ and standard deviation $s_i$.

$$\mu_i = \int_\Omega f_i(x) dx \tag{17}$$

$$\sigma_i^2 = \int_\Omega f_i(x)^2 dx - \mu_i^2 \tag{18}$$

$$g_i(x) = \frac{f_i(x) - \mu_i}{\sigma_i + \epsilon} \times s_i + m_i, \tag{19}$$

where $\epsilon > 0$ is a small constant. Just as in the discrete case, $\mu_i$ and $\sigma_i$ can be a moving average of the means and standard deviations observed over the course of training different NFs, and they can also be averaged over a minibatch of NFs (batch normalization) or calculated per datapoint (instance normalization). Mean $m_i$ and standard deviation $s_i$ can be learned directly (batch normalization), conditioned on other data (adaptive instance normalization), or fixed at 0 and 1 respectively (instance normalization).

**Multi-scale architectures**  Many discretizations permit multi-scale structures by subsampling the discretization, and QMC is particularly conducive to such design. Under QMC, downsampling is efficiently implemented by truncating the list of coordinates in the low discrepancy sequence to the desired number of terms, as the truncated sequence is itself low discrepancy. Similarly, upsampling can be implemented by extending the low discrepancy sequence to the desired number of terms, then performing interpolation by copying the nearest neighbor(s) or applying some (fixed or learned) kernel. Residual or skip connections can also be implemented efficiently since downsampling and upsampling are both specified with respect to the same discretization (Fig. 2 right).

### 5.4  Training DI-Nets

The pipeline for training DI-Nets is similar to that for training discrete networks, except that input and/or output discretizations should be specified. When training a DI-Net classifier, the input discretization may be specified manually or sampled from a low discrepancy sequence to perform QMC integration. The QMC discretization can be either deterministic or pseudorandom, and can accelerate computation when the same discretization is used for multiple network layers or all functions in a minibatch. When training DI-Nets for dense prediction (e.g. segmentation), the output discretization should be chosen to match the coordinates of the ground truth labels. Any input discretization can be chosen. Unless otherwise stated we set it equal to the output discretization. At inference time, the network can be evaluated with any output discretization, making the output in effect a neural field. We outline steps for training a classifier and dense prediction DI-Net in Algorithms 2 and 3 to illustrate their similarity to the discrete case, besides the specification of the discretization. Similar pipelines could be used to train DI-Net on other tasks including generative modeling, inverse problems, or representation learning. In most cases where discrete networks are sufficient, the gains for using DI-Nets may be limited.

## 5.5 Computational complexity

In general DI-Net's time and memory complexity both scale linearly with the number of sample points (regardless of the dimensionality of $\Omega$), as well as with network depth and width. Implemented naively, the computational cost of the continuous convolution is quadratic in the number of sample points, as it must calculate weights separately for each neighboring pair of points. But we can reduce this to a linear cost by specifying a $N_{\text{bin}}$ Voronoi partition of the kernel support $B$, then using the value of the kernel at each seed point for all points in its cell. Thus the kernel need only be evaluated $N_{\text{bin}}$ times regardless of the number of sample points. Additionally $N_{\text{bin}}$ can be modified during training and inference.

Thus DI-Nets are similar in computational complexity to discrete neural networks, with the added flexibility of being able to sample in non-grid patterns that can converge more efficiently. Although networks that operate directly on the parameters of the neural field can access the entire NF in constant time, the NF itself needs more parameters to capture finer resolutions, and in practice many downstream tasks can be solved at a much coarser resolution than would be captured by the NF. Thus such networks do not necessarily scale better than DI-Nets, and would likely suffer from higher computational costs when there is a mismatch between the resolution of the NF and the resolution needed for the task.

## 6 Experiments

We apply convolutional DI-Nets to toy classification (NF→scalar) and dense prediction (NF→NF) tasks, and analyze its behavior under different discretizations. Our aim is not to compete with discrete networks on these tasks, but rather to illustrate the learning behavior of CNNs compared to DI-Nets with the equivalent architectures, without additional techniques or types of layers. We demonstrate that convolutional DI-Nets learn maps that often generalize to unseen discretizations, whereas maps learned by pre-trained CNNs are highly sensitive to perturbations of the discretization.

### 6.1 Neural Field Classification

**Data**  We perform classification on a dataset of 8,400 NFs fit to a subset of ImageNet1k (Deng et al., 2009), with 700 samples from each of the 12 superclasses in the `big_12` dataset (Engstrom et al., 2019), which is derived from the WordNet hierarchy. We fit SIREN (Sitzmann et al., 2020b) to each image in ImageNet using 5 fully connected layers with 256 channels and sine non-linearities, trained for 2000 steps with an Adam optimizer at a learning rate of $10^{-4}$. It takes coordinates on $[-1, 1]^2$ and produces RGB values in $[-1, 1]^3$. We fit Gaussian Fourier feature (Tancik et al., 2020b) networks using 4 fully connected layers with 256 channels and ReLU activations. It takes coordinates on $[0, 1]^2$ and produces RGB values in $[0, 1]^3$. The average pixel-wise error of SIREN is $3 \cdot 10^{-4} \pm 2 \cdot 10^{-4}$, compared to $1.6 \cdot 10^{-2} \pm 8 \cdot 10^{-3}$ for Gaussian Fourier feature networks. The difference in quality is visible at high resolution, but indistinguishable at low resolution.

**Architecture and Baselines**  We train DI-Nets with 2 and 4 MLP convolutional layers, as well as CNNs with similar architectures. We also train an MLP that predicts class labels from SIREN parameters, and a "non-uniform convolution" (Jiang et al., 2019) that applies a non-uniform Fourier transform to input points (sampled with QMC) to map them to grid values, then applies a 2-layer CNN. DI-Net-2 uses strided MLP convolutions, a global average pooling layer, then two fully connected layers. DI-Net-4 adds a residual block with two MLP convolutions after the strided convolutions. We train all models with an AdamW optimizer (Loshchilov & Hutter, 2017). The architecture of the MLP is 3 fully connected layers with 128 hidden units each and ReLU activation separated by batch normalization. It learns to map the SIREN parameters to the class label. We found that the model's loss curve becomes unstable after 3000 iterations so we reduce the number of iterations to 2000. The non-uniform CNN applies the non-uniform Fourier transform (Muckley et al., 2020) followed by inverse Fast Fourier Transform to resample the input signal to the grid. It then feeds the result to a 2-layer CNN to perform classification.

**Training and Evaluation**  For each class, we train DI-Net on 500 SIRENs (Sitzmann et al., 2020b) and evaluate on 200 Gaussian Fourier feature networks (Tancik et al., 2020b). Training and testing on different

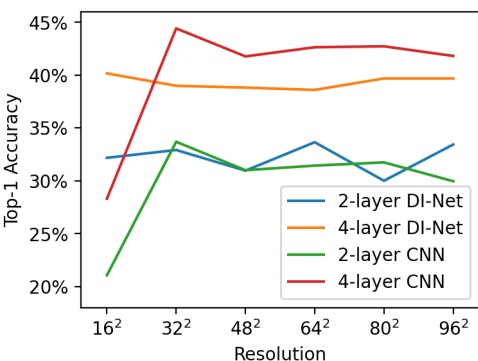

Figure 3: Classifier performance with different resolutions at test time.

NF types is not possible for the MLP approach, so it is evaluated on SIREN images instead. During training, we augment with noise, horizontal flips, and coordinate perturbations.

Each network is trained for 8K iterations with a learning rate of $10^{-3}$. In training, the CNNs sample neural fields along the $32 \times 32$ grid. DI-Nets and the non-uniform network sample 1024 points generated from a scrambled Sobol sequence (QMC discretization). We evaluate models with top-1 accuracy at the same resolution as well as at several other resolutions. A 4-layer DI-Net performs a forward pass on a batch of 48 images in $96 \pm 4$ms on a single NVIDIA RTX 2080 Ti GPU.

**Performance** The MLP and the non-uniform method significantly underperform DI-Net, with 13.9% and 28.9% accuracy respectively compared to 32.9% for our 2-layer network. At $32 \times 32$ resolution, DI-Nets somewhat underperform their CNN counterparts, and this performance gap is larger for deeper models. However, our discretization invariant model better generalizes to images of different resolutions than CNNs (Fig. 3), particularly at lower resolutions.

Table 1: Accuracy of 2-layer DI-Net under various discretizations.

| Train→Test Disc. | Accuracy |
|---|---|
| QMC→QMC | **32.9%** |
| Grid→Grid | 30.5% |
| Shrunk→Shrunk | 30.3% |
| QMC→Grid | 27.1% |
| Grid→QMC | 27.8% |
| QMC→Shrunk | 25.4% |
| Shrunk→QMC | 13.4% |

We next examine whether DI-Net can adapt to an entirely different type of discretization at test time. We use grid, QMC and shrunk discretizations of 1024 ($32 \times 32$) points. The Shrunk discretization shrinks a Sobol (QMC) sequence towards the center of the image (each point $x \in [-1, 1]^2$ is mapped to $x^2 \text{sgn}(x)$). In image classification, the object of interest is usually centered, hence the shrunk→shrunk setting performs on par with other discretizations despite its higher discrepancy.

Interestingly, changing discretization type at inference time has varying impact. Usually it only slightly degrades DI-Net's accuracy (Table 1), but performance falls dramatically when shifting from high to low discrepancy discretizations (shrunk→QMC). This observation points to the importance of training on the right discretizations to attain a network that generalizes well to other discretizations for the given task.

## 6.2 Neural Field Segmentation

**Data** We perform semantic segmentation of SIRENs fit to street view images from Cityscapes (Cordts et al., 2016), grouping segmentation labels into 7 categories. We train on 2975 NFs with coarsely annotated

segmentations only, and test on 500 NFs with both coarse and fine annotations (Fig. 4). We use a $48 \times 96$ grid discretization since segmentation labels are only given at pixel coordinates. SIREN is trained on Cityscapes images for 2500 steps, using the same architecture and settings as ImageNet. Seven segmentation classes are used for training and evaluation, labeled as 'flat' (e.g. road), 'construction' (e.g. building), 'object' (e.g. pole), 'nature', 'sky', 'human', and 'vehicle'.

**Architecture and Baselines**  We compare the performance of 3 and 5 layer DI-Nets and fully convolutional networks (FCNs), as well as a non-uniform CNN (Jiang et al., 2019). We also train a hypernetwork that learns to map each SIREN to the parameters of a new SIREN representing its segmentation.

DI-Net-3 uses two MLP convolutional layers at the same resolution followed by channel mixing (pointwise convolution). There are 16, 32 and 32 channels in the intermediate features. The support of the kernels in the MLP convolutional layers is $.025 \times .05$ and $.075 \times .15$ respectively, to account for the wide images in Cityscapes being remapped to $[-1, 1]^2$. DI-Net-5 uses a strided MLP convolution to perform downsampling and nearest neighbor interpolation for upsampling. There are 16 channels in all intermediate features. There is a residual connection between the higher resolution layers.

The CNN baselines use 3x3 convolutions with the same number of layers and channels as DI-Net. All networks use ReLU activation and batch normalization. The hypernetwork learns a map from the SIREN RGB to a SIREN with the same architecture that represents the segmentation. It predicts changes to the weights of all layers before the final fully connected layer, and predicts raw values for the weights of the final layer since it has 7 output channels for segmentation instead of 3 for RGB. The non-uniform CNN applies the non-uniform Fourier transform followed by inverse Fast Fourier Transform, and feeds the result to the 3-layer FCN to perform segmentation.

**Training and Evaluation**  Networks are trained for 10K iterations with a learning rate of $10^{-3}$. We evaluate each model with mean intersection over union (mIoU) and pixel-wise accuracy (PixAcc).

**Performance**  The hypernetwork and non-uniform CNN perform poorly compared to both FCNs and DI-Nets (Table 2). DI-Net-3 outperforms the equivalent FCN, and less often confuses features such as shadows and road markings (Fig. 4). However, the performance deteriorates when downsampling and upsampling layers are added (DI-Net-5), echoing the difficulty in scaling DI-Nets observed in classification. We suggest potential methods for remedying this in Section 7.

Table 2: Segmentation performance on NFs fit to Cityscapes images (trained on coarse segs).

| Model Type | Coarse Segs | | Fine Segs | |
|---|---|---|---|---|
| | mIoU | PixAcc | mIoU | PixAcc |
| 3-layer FCN | 0.409 | 69.6% | 0.374 | 63.6% |
| DI-Net-3 | **0.471** | **78.5%** | **0.417** | **69.4%** |
| 5-layer FCN | **0.488** | **79.4%** | **0.436** | **72.5%** |
| DI-Net-5 | 0.443 | 77.7% | 0.394 | 68.4% |
| Hypernetwork | 0.038 | 7.9% | 0.042 | 8.3% |
| Non-uniform | 0.109 | 26.5% | 0.106 | 22.7% |

### 6.3  Signed Distance Function Prediction

**Data**  We train a convolutional DI-Net to map a field of RGBA values in 3D to its signed distance function (SDF). We create a synthetic dataset of 3D scenes with randomly colored balls embedded in 3D space. Each toy scene contains 2-4 balls of random radii (range 0.2-0.5), centers, and colors scattered in 3D space ($\Omega = [-1, 1]^3$). For simplicity, we train each network directly on the closed form expressions for the RGBA fields and signed distance functions, rather than fitting neural fields first. We train on RGBA-SDF pairs using a mean squared error (MSE) loss on the predicted SDF. We use grid ($16 \times 16 \times 16$), QMC, shrunk, Monte Carlo (i.i.d. points drawn uniformly from the domain), and mixed discretizations of 4096 points. In the mixed setting, each minibatch uses one of the other four discretizations at random.

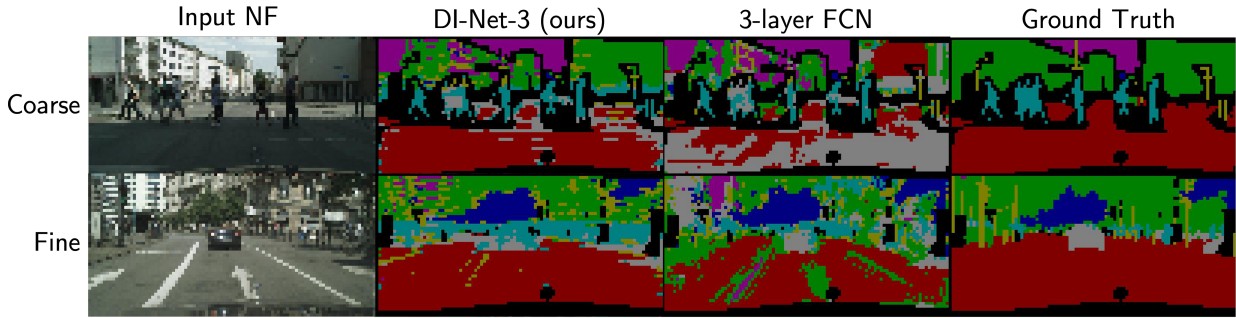

Figure 4: Cityscapes NF segmentations for models trained on coarse segmentations only. NF-Net produces NF segmentations, which can be evaluated at the subpixel level.

**Architecture and Baselines**  The FCN contains 3 convolutional layers of kernel lengths 3, 5 and 1 respectively. Accordingly, the convolutional DI-Net contained 2 convolutional layers followed by a linear combination layer. There are 8 channels in all intermediate features.

**Training and Evaluation**  We train each network for 1000 iterations with an AdamW optimizer with a batch size of 64 and a learning rate of 0.1 with an MSE loss on the SDF.

**Performance**  Under any fixed discretization, the convolutional DI-Net significantly outperforms the equivalent discrete network (MSE of 0.022 vs. 0.067 respectively). In Figure 5, we illustrate our model's ability to also produce outputs that are discretized differently than the input, making DI-Net's output in effect a neural field. By changing the output discretization of the last convolutional DI-Net layer, we can evaluate the output SDF anywhere on the domain without changing the input discretization. Whereas the discrete network is forced to output predictions at the resolution it was trained on, convolutional DI-Net can produce outputs along a high-quality grid discretization given a coarse QMC discretization, even when it is only trained under QMC output discretizations.

Table 3: Mean squared error ($\times 10^{-2}$) of predicted SDFs under different discretizations. The three best settings are bolded. MC=Monte Carlo.

| Train \ Test | Grid | QMC | Shrunk | MC |
|---|---|---|---|---|
| Grid | **2.18** | 2.54 | 3.77 | 3.81 |
| QMC | 3.60 | **2.01** | 2.94 | 3.92 |
| Shrunk | 3.72 | 2.88 | **2.00** | 4.30 |
| MC | 6.45 | 5.97 | 4.89 | 5.92 |
| Mixed | 4.65 | 4.41 | 3.26 | 4.09 |

DI-Net performs almost equally well under grid, QMC and shrunk discretizations in the in-distribution setting, but on this task it is more sensitive to out-of-distribution discretizations than the classifier in Section 6.1. While shrunk→QMC is the worst performing combination for the classifier, here it is one of the better performing combinations. DI-Net likely struggles with Monte Carlo sampling due to its high discrepancy discretizations, resulting in cases where smaller balls are entirely missed. Interestingly, the model fares worse when trained on multiple discretizations simultaneously, suggesting that the network may be guided in opposing directions by different discretizations resulting in unstable training. These observations illustrate the complex task-dependent interplay between the type of discretizations observed at training time and the ability of the model to generalize to new discretizations.

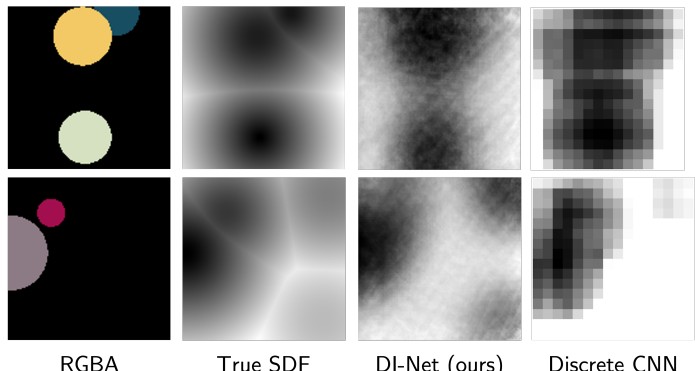

| RGBA | True SDF | DI-Net (ours) | Discrete CNN |

Figure 5: 2D slices of two toy 3D scenes (black is transparent) with signed distance functions predicted by DI-Net and a fully convolutional network.

## 6.4   Initialization with discrete networks

A convolutional DI-Net can be initialized with the weights of a pre-trained convolutional neural network. In fact, every CNN induces a convolutional DI-Net that behaves identically when every layer is restricted to a particular grid discretization. However, the behavior of the pre-trained CNN is not preserved when the DI-Net switches to other discretizations. Tiny perturbations from the regular grid can change a classifier's predictions, as small differences in each layer can accumulate to exert a large influence on the final output. Figure 6 illustrates this phenomenon for a DI-Net initialized with a truncated EfficientNet.

Moreover, we find that once grid discretization is abandoned, large DI-Nets cannot easily be fine-tuned to restore the behavior of the discrete network used to initialize it. In Tables 4 and 5, we illustrate that DI-Net initialized with a large pre-trained discrete network does not match the performance of the original model when fine-tuned with QMC sampling. We use a truncated version of EfficientNet (Tan & Le, 2019) for classification, and fine-tune for 200 samples per class. For segmentation we use a truncated version of ConvNexT-UPerNet (Liu et al., 2022), fine-tuning with 1000 samples.

Table 4: Pre-trained models fine-tuned on ImageNet NF classification.

| Model Type | Accuracy |
|---|---|
| EfficientNet (Tan & Le, 2019) | **66.4%** |
| DI-Net-EN | 48.1% |

Table 5: Pre-trained models fine-tuned on Cityscapes segmentation.

| Model Type | Mean IoU | Pixel Accuracy |
|---|---|---|
| ConvNexT (Liu et al., 2022) | **0.429** | 68.1% |
| DI-Net-CN | 0.376 | **68.7%** |

These observations suggest that the optimization landscape of discretized maps can vary significantly with small changes in $X$, and that the behavior of pre-trained CNNs should not be expected to generalize well to other discretizations. We suspect that this instability may be linked to the innate sensitivity of the grid discretization due (at least in part) to its higher discrepancy. We observe that the output of a randomly initialized DI-Net is less stable under changing sampling resolution with a grid pattern relative to a low discrepancy discretization (Fig. 7). While the output of a DI-Net with QMC sampling converges at high resolution, the grid sampling scheme has unstable outputs until very high resolution. Only the grids that overlap each other (resolutions in powers of two) produce similar activations.

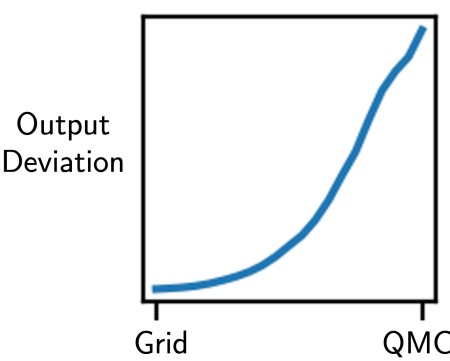
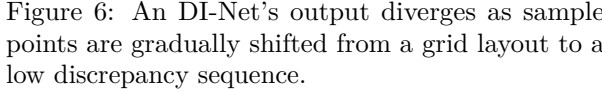

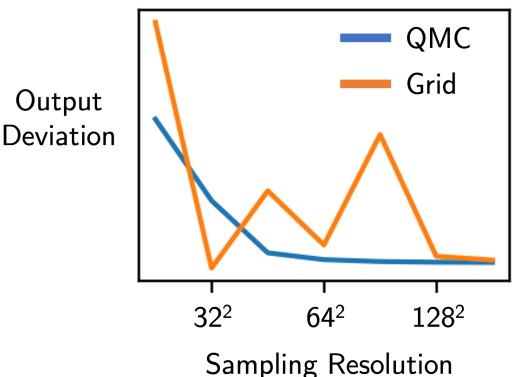

Figure 6: An DI-Net's output diverges as sample points are gradually shifted from a grid layout to a low discrepancy sequence.

Figure 7: Distance of the output of a DI-Net from its grid output at $32 \times 32$ resolution, when sampling at various resolutions. Its outputs deviate rapidly as its discretization shifts from a regular grid to a low discrepancy sequence

## 7 Future Directions

**Discretization invariance as learning signal**   Our experiments show that a gap remains in a trained DI-Net's ability to generalize to unseen discretizations at test time. One approach to bridging this generalization gap may be to explicitly encourage discretization invariance during training by adding a regularization term that minimizes the distance between intermediate features obtained with different discretizations. Such a scheme could also be used for contrastive pre-training, where different discretizations of the same neural field should be projected onto closer points in latent space than discretizations of different NFs.

**Extending DI-Net to adaptive, high discrepancy discretizations**   Our approach of choosing the discretization of the domain *a priori* breaks down in applications where large regions of the domain are less informative for the task of interest. For example, most of the information in 3D scenes is concentrated at object surfaces, so densely sampling all five dimensions of a neural radiance field is a poor choice of discretization for most downstream tasks. Moreover, ground truth labels for dense prediction tasks may only be available along a high discrepancy discretization. However, low discrepancy sampling can still guide an initial discretization (e.g. selecting an initial set of camera poses or rays), and adaptive sampling techniques such as adaptive quadrature or sphere tracing can be used to refine the discretization. Future work should explore the design of DI-Nets that can achieve low approximation errors under task-specific discretizations. It may also be possible to reduce the number of samples needed to achieve low discretization error over the course of training or at inference time, for example by learning which discretizations produce more reliable estimates, or by designing layers that encourage the integral to converge to a predefined set of quantized values and propagating these quantized values downstream.

**Error propagation**   Neural fields do not always faithfully represent the underlying data, whether due to insufficient coverage by sensors or due to a suboptimally trained field. Our analysis can be extended to account for this influence on the error in a model's output. In the worst case, these deviations are adversarial examples, and robustness techniques for discrete networks can also be applied to DI-Net. But what can we say about typical deviations of NFs? Future work should analyze patterns in the mistakes that different types of NFs make, and investigate how to make downstream models robust to these.

## 8 Conclusion

We present a general framework for understanding and constructing discretization invariant neural networks, highlighting the importance of discrepancy in bounding the deviation of the network outputs under differ-

ent discretizations, as well as establishing convergence in both its learning and inference behavior under equidistributed discretization sequences. DI-Nets can learn arbitrary maps between integrable functions of bounded variation, making them a useful tool for performing inference on neural fields in a parameterization-agnostic manner, or for applications to systems that process a continuous signal by querying it on a point set. Discretization invariance may be a particularly useful concept to harness in the context of 3D scene understanding, as the information in a scene that is relevant for most tasks of interest is invariant under a much wider range of discretizations (e.g., points, rays and light fields; 360 degree or forward-facing) than can be described purely by group symmetries. With the increasing popularity and diversity of neural fields as well as the emergence of tools to efficiently create large datasets of NFs, understanding discretization invariant learning may be key to developing interoperable approaches for learning on such data.

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

# Appendix

Appendix A provides additional background on the variation of a function and discrepancy of a point set, as well as more general forms of DI layers. Appendix B provides detailed proofs of the Universal Approximation and Convergent Empirical Gradients theorems, as well as some corollaries. Appendix C provides additional examples of DI layers that extend convolutional neural networks and vision transformers.

## A  More Details on Discretization Invariance

### A.1  Koksma–Hlawka inequality and low discrepancy sequences

Recall that a function $f \in L^2(\Omega)$ satisfies a Koksma–Hlawka inequality if for any point set $X \subset \Omega$,

$$\left| \frac{1}{|X|} \sum_{x' \in X} f(x') - \int_\Omega f(x)\, dx \right| \leq V(f)\, D(X), \tag{20}$$

for normalized measure $dx$, some notion of variation $V$ of the function and some notion of discrepancy $D$ of the point set. The classical inequality gives a tight error bound for functions of bounded variation in the sense of Hardy-Krause (BVHK), a generalization of bounded variation to multivariate functions on $[0,1]^d$ which has bounded variation in each variable. Specifically, the variation is defined as:

$$V_{HK}(f) = \sum_{\alpha \in \{0,1\}^d} \int_{[0,1]^{|\alpha|}} \left| \frac{\partial^\alpha}{\partial x^\alpha} f(x_\alpha) \right| dx, \tag{21}$$

with $\{0,1\}^d$ the multi-indices and $x_\alpha \in [0,1]^d$ such that $x_{\alpha,j} = x_j$ if $j \in \alpha$ and $x_{\alpha,j} = 1$ otherwise. The classical inequality also uses the star discrepancy of the point set $X$, given by:

$$D^*(X) = \sup_{I \in J} \left| \frac{1}{|X|} \sum_{x' \in X} \mathbb{1}_I(x_j) - \lambda(I) \right|, \tag{22}$$

where $J$ is the set of $d$-dimensional intervals that include the origin, and $\lambda$ the Lebesgue measure.

A point set is called low discrepancy if its discrepancy is on the order of $O((\ln N)^d/N)$. Quasi-Monte Carlo calculates the sample mean using a low discrepancy sequence (see Fig. A.1 for examples in 2D), as opposed to the i.i.d. point set generated by standard Monte Carlo, which will generally be high discrepancy. Because the Koksma–Hlawka inequality is sharp, when estimating the integral of a BVHK function on $[0,1]^d$, the error of the QMC approximation decays as $O((\ln N)^d/N)$, in contrast to the error of the standard Monte Carlo approximation that decays as $O(N^{-1/2})$ (Caflisch, 1998).

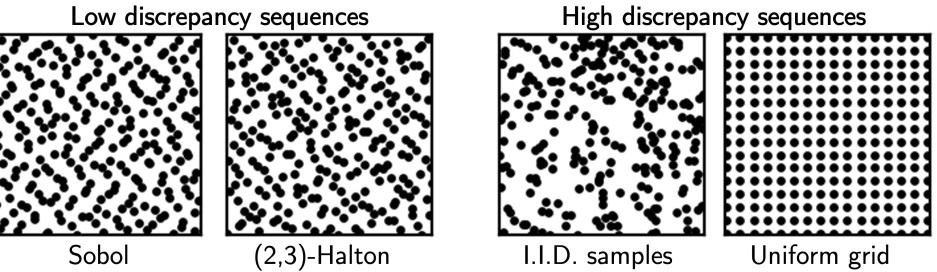

Figure A.1: Examples of low and high discrepancy sequences in 2D.

However, BVHK is a rather restrictive class of functions defined on $[0,1]^d$ that excludes all functions with discontinuities. Brandolini et al. (2013) extended the Koksma–Hlawka inequality to two classes of functions defined below:

**Piecewise smooth functions**  Let $f$ be a smooth function on $[0,1]^d$ and $\Omega$ a Borel subset of $[0,1]^d$. Then $f|_\Omega$ is a piecewise smooth function with the Koksma–Hlawka inequality given by variation

$$V(f) = \sum_{\alpha \in \{0,1\}^d} 2^{d-|\alpha|} \int_{[0,1]^d} \left| \frac{\partial^\alpha}{\partial x^\alpha} f(x) \right| dx, \tag{23}$$

and discrepancy:

$$D(X) = 2^d \sup_{I \subseteq [0,1]^d} \left| \frac{1}{|X|} \sum_{x' \in X} \mathbb{1}_{\Omega \cap I}(x_j) - \lambda(\Omega \cap I) \right|. \tag{24}$$

**$W^{d,1}$ functions on manifolds**  Let $\mathscr{M}$ be a smooth compact $d$-dimensional manifold with normalized measure $dx$. Given local charts $\{\phi_k\}_{k=1}^K$, $\phi_k : [0,1]^d \to \mathscr{M}$, the variation of a function $f \in W^{d,1}(\mathscr{M})$ is characterized as:

$$V(f) = c \sum_{k=1}^K \sum_{|\alpha| \leq n} \int_{[0,1]^d} \left| \frac{\partial^\alpha}{\partial x^\alpha} (\psi_k(\phi_k(x)) f(\phi_k(x))) \right| dx, \tag{25}$$

with $\{\psi_k\}_{k=1}^K$ a smooth partition of unity subordinate to the charts, and constant $c > 0$ that depends on the charts but not on $f$. Defining the set of intervals in $\mathscr{M}$ as $J = \{U : U = \phi_k(I) \text{ for some } k \text{ and } I \subseteq [0,1]^d\}$, with measure $\mu(U) = \lambda(I)$, the discrepancy of a point set $Y = \{y_j\}_{y=1}^N$ on $\mathscr{M}$ is:

$$D(Y) = \sup_{U \in J} \left| \frac{1}{|X|} \sum_{x' \in X} \mathbb{1}_U(y_j) - \mu(U) \right|. \tag{26}$$

**Neural fields**  We define the variation of a neural field as the sum of the variations of each channel.

**Note**: The notion of discrepancy is not limited to the Lebesgue measure. The existence of low discrepancy point sets has been proven for non-negative, normalized Borel measures on $[0,1]^d$ due to Aistleitner & Dick (2013). An extension of our framework to non-uniform measures is a promising direction for future work (see Appendix 7).

### A.2  More General forms of DI layers

Recall that we defined DI layers as having the form $\mathcal{H}_\phi[f] = \int_\Omega H_\phi[f](x)dx$ for neural fields $f$ (we drop $\theta$ here for readability). In the case where $H_\phi[f] : \Omega \to \mathbb{R}^n$, i.e. $\mathcal{H}_\phi$ is a layer that maps neural fields to vectors, we permit layers of the following more general form:

$$\mathcal{H}_\phi[f] = \int_\Omega h_\phi(x, f(x), \dots, D^\alpha f(x)) d\mu(x), \tag{27}$$

for weak derivatives up to order $|\alpha|$ taken with respect to each channel. $D^\alpha f = \frac{\partial^{|\alpha|} f}{\partial x_1^{\alpha_1} \dots \partial x_n^{\alpha_n}}$ for multi-index $\alpha$. The dependence of $h$ on weak derivatives up to order $k = |\alpha|$ requires that the weak derivatives are integrable, i.e., $f$ is in the Sobolev space $W^{k,2}(\Omega)$, and that $h_\phi$ is Gateaux differentiable w.r.t. these weak derivatives. Note that a non-uniform measure $\mu$ changes the discrepancy of sampled sequences.

In the function-valued case, we can similarly have:

$$\mathcal{H}_\phi[f](x') = \int_\Omega H_\phi[f](x, x') d\mu(x) \tag{28}$$

$$= \int_\Omega h_\phi(x, f(x), \dots, D^\alpha f(x), x', f(x'), \dots, D^\alpha f(x')) d\mu(x), \tag{29}$$

where we require $H_\phi[f](\cdot, x') \in \mathcal{F}_n$. Both of our key theoretical results (convergence of discretized gradients and universal approximation) apply to this general form. Gateaux differentiability of $h_\phi$ allows us to apply the same proof in Appendix B.2 to the derivatives. Since allowing layers to depend on weak derivatives results in an even more expressive class of DI-Nets, the universal approximation theorem still holds.

# B  Proofs

## B.1  Proof of Lemma 3.2 (convergence of discretized gradients w.r.t. inputs)

*Proof.* Here we combine the function to vector and function to function cases for brevity. For fixed $\tilde{x} \in \Omega$, the discretized derivative of $\mathcal{H}_\phi$ w.r.t. $f(\tilde{x})$ can be written:

$$\frac{\partial}{\partial f(\tilde{x})} \hat{\mathcal{H}}_\phi^N[f] = \frac{\partial}{\partial f(\tilde{x})} \left( \frac{1}{|X_N|} \sum_{x \in X_N} H_\phi[f](x) \right) \tag{30}$$

$$= \lim_{\tau \to 0} \frac{1}{\tau |X_N|} \sum_{x \in X_N} H_\phi[f + \tau \psi_{\tilde{x}}^N](x) - H_\phi[f](x), \tag{31}$$

where $\psi_{\tilde{x}}^N$ is any function in $W^{|\alpha|,1}(\Omega)$ that is 1 at $\tilde{x}$ and 0 on $X_N \backslash \{\tilde{x}\}$, and whose derivatives are 0 on $X_N$. As an example, take the bump function which vanishes outside a small neighborhood of $\tilde{x}$ and smoothly ramps to 1 on a smaller neighborhood of $\tilde{x}$, making its weak derivative 0 at $\tilde{x}$.

By (3) we know that the sequences $\left\| \hat{\mathcal{H}}_\phi^N[f] - \mathcal{H}_\phi[f] \right\|$ and $\left\| \hat{\mathcal{H}}_\phi^N[f + \tau \psi_{\tilde{x}}^N] - \mathcal{H}_\phi[f + \tau \psi_{\tilde{x}}^N] \right\|$ converge uniformly in $N$ to 0 for any $\tau > 0$, where we can use the $\ell_1$ norm for vector outputs or the $L^1$ norm for function outputs. So for any $\epsilon > 0$ and any $\tau > 0$, we can choose $N_0$ large enough such that for any $N > N_0$:

$$\left\| \hat{\mathcal{H}}_\phi^N[f + \tau \psi_{\tilde{x}}^N] - \mathcal{H}_\phi[f + \tau \psi_{\tilde{x}}^N] \right\| < \frac{\epsilon}{2}, \tag{32}$$

and

$$\left\| \hat{\mathcal{H}}_\phi^N[f] - \mathcal{H}_\phi[f] \right\| < \frac{\epsilon}{2}. \tag{33}$$

Then,

$$\left\| \hat{\mathcal{H}}_\phi^N[f + \tau \psi_{\tilde{x}}^N] - \mathcal{H}_\phi[f + \tau \psi_{\tilde{x}}^N] \right\| + \left\| \hat{\mathcal{H}}_\phi^N[f] - \mathcal{H}_\phi[f] \right\| < \epsilon, \tag{34}$$

by the triangle inequality,

$$\left\| (\hat{\mathcal{H}}_\phi^N[f + \tau \psi_{\tilde{x}}^N] - \mathcal{H}_\phi[f + \tau \psi_{\tilde{x}}^N]) - (\hat{\mathcal{H}}_\phi^N[f] - \mathcal{H}_\phi[f]) \right\| < \epsilon \tag{35}$$

$$\left\| \left( \hat{\mathcal{H}}_\phi^N[f + \tau \psi_{\tilde{x}}^N] - \hat{\mathcal{H}}_\phi^N[f] \right) - \left( \mathcal{H}_\phi[f + \tau \psi_{\tilde{x}}^N] - \mathcal{H}_\phi[f] \right) \right\| < \epsilon, \tag{36}$$

hence $\left\| \left( \hat{\mathcal{H}}_\phi^N[f + \tau \psi_{\tilde{x}}^N] - \hat{\mathcal{H}}_\phi^N[f] \right) - \left( \mathcal{H}_\phi[f + \tau \psi_{\tilde{x}}^N] - \mathcal{H}_\phi[f] \right) \right\|$ converges uniformly to 0. Since the distance between two vectors is 0 iff they are the same, we can write:

$$\lim_{N \to \infty} \hat{\mathcal{H}}_\phi^N[f + \tau \psi_{\tilde{x}}^N] - \hat{\mathcal{H}}_\phi^N[f] = \lim_{N \to \infty} \mathcal{H}_\phi[f + \tau \psi_{\tilde{x}}^N] - \mathcal{H}_\phi[f] \tag{37}$$

$$\lim_{\tau \to 0} \frac{1}{\tau} \lim_{N \to \infty} \left( \hat{\mathcal{H}}_\phi^N[f + \tau \psi_{\tilde{x}}^N] - \hat{\mathcal{H}}_\phi^N[f] \right) = \lim_{\tau \to 0} \frac{1}{\tau} \lim_{N \to \infty} \left( \mathcal{H}_\phi[f + \tau \psi_{\tilde{x}}^N] - \mathcal{H}_\phi[f] \right). \tag{38}$$

By the Moore-Osgood theorem,

$$\lim_{N \to \infty} \lim_{\tau \to 0} \frac{1}{\tau} \left( \hat{\mathcal{H}}_\phi^N[f + \tau \psi_{\tilde{x}}^N] - \hat{\mathcal{H}}_\phi^N[f] \right) = \lim_{N \to \infty} \lim_{\tau \to 0} \frac{1}{\tau} \left( \mathcal{H}_\phi[f + \tau \psi_{\tilde{x}}^N] - \mathcal{H}_\phi[f] \right) \tag{39}$$

$$\lim_{N \to \infty} \frac{\partial}{\partial f(\tilde{x})} \hat{\mathcal{H}}_\phi^N[f] = \lim_{N \to \infty} d\mathcal{H}_\phi[f; \psi_{\tilde{x}}^N]. \tag{40}$$

Since $h_\phi$ is Gateaux differentiable and bounded, $H_\phi$ is also Gateaux differentiable for $f$ of bounded variation, hence the limit on the right hand side is finite.

For each discretization $X_N$, choose a sequence of bump functions around each $x \in X_N$, $\{\psi_x^N\}_{x \in X_N}$. An example of such a family is the (appropriately designed) partitions of unity with $|X_N|$ elements.

Then the discretized gradient converges to the limit of the Gateaux derivatives of $\mathcal{H}_\phi$ w.r.t. the bump function sequence as $N \to \infty$. $\qquad\square$

## B.2 Proof of Theorem 3.3 (convergence of discretized gradients)

*A DI-Net permits backpropagation with respect to its input and all its learnable parameters. The discretized gradients converge under any equidistributed discretization sequence.*

We note that this property automatically holds if the layer does not perform numerical integration. This includes layers which take $\mathbb{R}^n$ as input, as well as point-wise transformations. Then the (sub)derivatives with respect to inputs and parameters need only be well-defined at each point of the output in order to enable backpropagation.

Choose an equidistributed discretization sequence $\{X_N\}_{N\in\mathbb{N}}$ on $\Omega$. We consider a DI layer $\mathcal{H}_\phi$ which takes a function $f$ as input and may output a vector or function.

$$\mathcal{H}_\phi[f] = \int_\Omega H_\phi[f](x)dx. \tag{41}$$

Recall its discrete operator under $X_N$:

$$\hat{\mathcal{H}}_\phi^N[f] = \frac{1}{|X_N|} \sum_{x \in X_N} H_\phi[f](x). \tag{42}$$

**Lemma B.1.** *Chained discretized derivatives converge to the chained Gateaux derivatives.*

*Proof.* Consider a two-layer DI-Net with function input $f \mapsto (\mathcal{H}_\theta \circ \mathcal{H}_\phi)[f]$. For the case of derivatives w.r.t. the input, we would like to show the analogue of (40):

$$\lim_{N\to\infty} \frac{\partial}{\partial f(\tilde{x})} \left( \hat{\mathcal{H}}_\theta^N \circ \hat{\mathcal{H}}_\phi^N \right) [f] = \lim_{N\to\infty} d \left( \mathcal{H}_\theta \circ \mathcal{H}_\phi \right) [f; \psi_{\tilde{x}}^N], \tag{43}$$

where the bump function $\psi_{\tilde{x}}^N$ is defined similarly (1 at $\tilde{x}$ and 0 at each $x \neq \tilde{x}$).

$$\frac{\partial}{\partial f(\tilde{x})} \left( \hat{\mathcal{H}}_\theta^N \circ \hat{\mathcal{H}}_\phi^N \right) [f] = \frac{\partial}{\partial f(x)} \left( H_\theta \left[ \hat{\mathcal{H}}_\phi^N[f] \right] (x) \right) \tag{44}$$

$$= \lim_{\tau\to0} \frac{1}{\tau} \left( H_\theta \left[ \hat{\mathcal{H}}_\phi^N[f + \tau\psi_{\tilde{x}}^N] \right] (x) - H_\theta \left[ \hat{\mathcal{H}}_\phi^N[f] \right] (x) \right) \tag{45}$$

as in (31).

By (3) we know $\left\| \mathcal{H}_\theta \left[ \hat{\mathcal{H}}_\phi^N[f + \tau\psi_{\tilde{x}}^N] \right] - \hat{\mathcal{H}}_\theta^N \left[ \hat{\mathcal{H}}_\phi^N[f + \tau\psi_{\tilde{x}}^N] \right] \right\|$ converges to 0 in $N$ for all $\tau > 0$ (where we can use the $\ell_1$ norm for vector outputs or the $L^1$ norm for function outputs), as does $\left\| \mathcal{H}_\theta \left[ \hat{\mathcal{H}}_\phi^N[f] \right] - \hat{\mathcal{H}}_\theta^N \left[ \hat{\mathcal{H}}_\phi^N[f] \right] \right\|_{L^1}$. Reasoning as in (32)-(39), we have:

$$\lim_{N\to\infty} \lim_{\tau\to0} \frac{1}{\tau} \left( \hat{\mathcal{H}}_\theta^N \left[ \hat{\mathcal{H}}_\phi^N[f + \tau\psi_{\tilde{x}}^N] \right] - \hat{\mathcal{H}}_\theta^N \left[ \hat{\mathcal{H}}_\phi^N[f] \right] \right) \tag{46}$$

$$= \lim_{\tau\to0} \frac{1}{\tau} \lim_{N\to\infty} \left( \mathcal{H}_\theta \left[ \hat{\mathcal{H}}_\phi^N[f + \tau\psi_{\tilde{x}}^N] \right] - \mathcal{H}_\theta \left[ \hat{\mathcal{H}}_\phi^N[f] \right] \right) \tag{47}$$

$$= \lim_{N\to\infty} \lim_{\tau\to0} \frac{1}{\tau} \left( \mathcal{H}_\theta \left[ \hat{\mathcal{H}}_\phi^N[f + \tau\psi_{\tilde{x}}^N] \right] - \mathcal{H}_\theta \left[ \hat{\mathcal{H}}_\phi^N[f] \right] \right) \tag{48}$$

Note that

$$d\mathcal{H}_\phi[f; \psi_{\tilde{x}}^N] = \frac{1}{\tau} \left( \mathcal{H}_\phi[f + \tau\psi_{\tilde{x}}^N] - \mathcal{H}_\phi[f] + o(\tau) \right) \tag{49}$$

$$\mathcal{H}_\phi[f + \tau\psi_{\tilde{x}}^N] = \mathcal{H}_\phi[f] + \tau d\mathcal{H}_\phi[f; \psi_{\tilde{x}}^N] + o(\tau). \tag{50}$$

Then we complete the equality in (43) as follows:

$$\text{LHS} = \lim_{N \to \infty} \frac{\partial}{\partial f(\tilde{x})} \left( \hat{\mathcal{H}}_\theta^N \circ \hat{\mathcal{H}}_\phi^N \right) [f] \tag{51}$$

$$= \lim_{N \to \infty} \lim_{\tau \to 0} \frac{1}{\tau} \left( \mathcal{H}_\theta \left[ \hat{\mathcal{H}}_\phi^N [f + \tau \psi_{\tilde{x}}^N] \right] - \mathcal{H}_\theta \left[ \hat{\mathcal{H}}_\phi^N [f] \right] \right) \tag{52}$$

$$= \lim_{N \to \infty} \lim_{\tau \to 0} \frac{1}{\tau} \left( \mathcal{H}_\theta \left[ \mathcal{H}_\phi [f] + \tau d\mathcal{H}_\phi [f; \psi_{\tilde{x}}^N] \right] - \mathcal{H}_\theta \left[ \mathcal{H}_\phi [f] \right] \right) \tag{53}$$

$$= \lim_{N \to \infty} d\mathcal{H}_\theta \left[ \mathcal{H}_\phi [f]; d\mathcal{H}_\phi [f; \psi_{\tilde{x}}^N] \right] \tag{54}$$

$$= \lim_{N \to \infty} d \left( \mathcal{H}_\theta \circ \mathcal{H}_\phi \right) [f; \psi_{\tilde{x}}^N] \tag{55}$$

$$= \text{RHS}, \tag{56}$$

by the chain rule for Gateaux derivatives.

The case of derivatives w.r.t. parameters is straightforward. In the same way we used (32)-(39) to obtain (48), we have:

$$\lim_{N \to \infty} \frac{\partial}{\partial \phi_k} (\hat{\mathcal{H}}_\theta^N \circ \hat{\mathcal{H}}_\phi^N)[f] = \lim_{N \to \infty} \lim_{\tau \to 0} \frac{1}{\tau} \left( \hat{\mathcal{H}}_\theta^N \left[ \hat{\mathcal{H}}_{\phi + \tau e_k}^N [f] \right] - \hat{\mathcal{H}}_\theta^N \left[ \hat{\mathcal{H}}_\phi^N [f] \right] \right) \tag{57}$$

$$= \lim_{\tau \to 0} \frac{1}{\tau} \left( \mathcal{H}_\theta [\mathcal{H}_{\phi + \tau e_k} [f]] - \mathcal{H}_\theta [\mathcal{H}_\phi [f]] \right) \tag{58}$$

$$= \frac{\partial}{\partial \phi_k} (\mathcal{H}_\theta \circ \mathcal{H}_\phi)[f], \tag{59}$$

By induction, the chained derivatives converge for an arbitrary number of layers. □

Since the properties of DI-Net layers extend to loss functions on DI-Nets, we can treat a loss function similarly to a layer. For a loss on model output $g$ and optional ground truth label $g'$:

$$\mathcal{L}_{g'}[g] = \int_\Omega L[g, g'](x)dx, \tag{60}$$

with discrete operator:

$$\hat{\mathcal{L}}_{g'}^N[g] = \frac{1}{|X_N|} \sum_{x \in X_N} L[g, g'](x), \tag{61}$$

we can write:

$$\lim_{N \to \infty} \frac{\partial}{\partial f(\tilde{x})} \left( \hat{\mathcal{L}}_{g'}^N \circ \hat{\mathcal{H}}_\theta^N \circ \hat{\mathcal{H}}_\phi^N \right) [f] = \lim_{N \to \infty} d \left( \mathcal{L}_{g'} \circ \mathcal{H}_\theta \circ \mathcal{H}_\phi \right) [f; \psi_{\tilde{x}}^N]. \tag{62}$$

So by Lemma 3.2 and B.1, we have shown that backpropagation is discretization invariant.

### B.3 Proof of Theorem 4.1 (Universal Approximation Theorem)

**Note**: By our definition of $\mathcal{F}_c$ (Section 5.1), there exists $V^*$ such that every $f \in \mathcal{F}_1$ satisfies a Koksma–Hlawka inequality (3) with $V(|f|) < V^*$.

$\mathcal{F}_1$ is bounded in $L^1$ norm since all their functions are compactly supported and bounded.

Consider a Lipschitz continuous map $\mathcal{R} : \mathcal{F}_1 \to \mathcal{F}_1$ such that $d(\mathcal{R}[f], \mathcal{R}[g])_{L^1} \le M_0 d(f, g)_{L^1}$ for some constant $M_0$ and all $f, g \in \mathcal{F}_1$. Let $M = \max\{M_0, 1\}$.

Fix a discretization $X \subset \Omega$ with discrepancy $D(X) = \frac{\epsilon}{12(M+2)V^*}$. By (3) this yields:

$$\left| \frac{1}{|X|} \sum_{x' \in X} f(x') - \int_\Omega f(x) \, dx \right| \le \frac{\epsilon}{12(M+2)}, \tag{63}$$

for all $f \in \mathcal{F}_1$. Let $N$ be the number of points in $X$.

**Definition B.2.** For given discretization $X$, the projection $\pi : f \mapsto \mathbf{f}$ is a quotient map $L^2(\Omega) \to L^2(\Omega)/\sim$ under the equivalence relation $f \sim g$ iff $f(x) = g(x)$ for all $x \in X$.

$L^2(\Omega)/\sim$ is isomorphic to $\mathbb{R}^{|X|}$, and thus can be given the normalized $\ell^1$ norm:

$$\|\pi f\|_{\ell^1} = \frac{1}{|X|} \sum_{x' \in X} |f(x')|. \tag{64}$$

**Definition B.3.** Denote the preimage of $\pi$ as $\boldsymbol{\pi}^{-1} : \mathbf{f}' \mapsto \{f' \in \mathcal{F}_1 : \pi f' = \mathbf{f}'\}$. Invoking the axiom of choice, define the inverse projection $\pi^{-1} : \pi\mathcal{F}_1 \to \mathcal{F}_1$ by a choice function over the sets $\boldsymbol{\pi}^{-1}(\pi\mathcal{F}_1)$.

Note that this inverse projection corresponds to some way of interpolating the $N$ sample points such that the output is in $\mathcal{F}_1$. Although our definition implies the existence of such an interpolator, we leave its specification as an open problem. Since $\Omega$ only permits discontinuities along a fixed Borel subset of $[0,1]^d$, these boundaries can be specified *a priori* in the interpolator. Since all functions in $\mathcal{F}_1$ are bounded and continuous outside this set, the interpolator can be represented by a bounded continuous map, hence it is expressible by a DI-Net layer.

**Definition B.4.** $\pi$ generates a $\sigma$-algebra on $\mathcal{F}_1$ given by $\mathscr{A} = \{\boldsymbol{\pi}^{-1}(S) : S \in \mathscr{L}\}$, with $\mathscr{L}$ the $\sigma$-algebra of Lebesgue measurable sets on $\mathbb{R}^N$. Because this $\sigma$-algebra depends on $\epsilon$ and the Lipschitz constant of $\mathcal{R}$ via the point set's discrepancy, we may write it as $\mathscr{A}_{\epsilon,\mathcal{R}}$.

In this formulation, we let the tolerance $\epsilon$ and the Lipschitz constant of $\mathcal{R}$ dictate what subsets of $\mathcal{F}_1$ are measurable, and thus which measures on $\mathcal{F}_1$ are permitted. However, if the desired measure $\nu$ is more fine-grained than what is permitted by $\mathscr{A}_{\epsilon,\mathcal{R}}$, then it is $\nu$ that should determine the number of sample points $N$, rather than $\epsilon$ or $\mathcal{R}$.

We now state the following lemmas which will be used to prove our universal approximation theorem.

**Lemma B.5.** *There is a map $\tilde{\mathcal{R}} : \pi\mathcal{F}_1 \to \pi\mathcal{F}_1$ such that*

$$\int_\Omega \left| \mathcal{R}[f](x) - \pi^{-1} \circ \tilde{\mathcal{R}} \circ \pi[f](x) \right| \, dx = \frac{\epsilon}{6}. \tag{65}$$

*Proof.* Let $g(x) = |f(x)|$ for $f \in \mathcal{F}_1$. Because (63) applies to $g(x)$, we have:

$$\left| \frac{1}{|X|} \sum_{x' \in X} g(x') - \int_\Omega g(x) \, dx \right| \le \frac{\epsilon}{12(M+2)} \tag{66}$$

$$\left| \|\pi f\|_{\ell^1} - \|f\|_{L^1} \right| \le \frac{\epsilon}{12(M+2)}. \tag{67}$$

Eqn. (67) also implies that for any $\mathbf{f} \in \pi\mathcal{F}_1$, we have:

$$\left| \|\mathbf{f}\|_{\ell^1} - \left\|\pi^{-1}\mathbf{f}\right\|_{L^1} \right| \le \frac{\epsilon}{12(M+2)}. \tag{68}$$

Combining (67) and (68), we obtain

$$\left| \|f\|_{L^1} - \left\|\pi^{-1} \circ \pi[f]\right\|_{L^1} \right| \le \frac{\epsilon}{6(M+2)}. \tag{69}$$

By the triangle inequality and applying $\mathcal{R}$:

$$\int_\Omega \left| \mathcal{R}[f](x) - \pi^{-1} \circ \pi \circ \mathcal{R}[f](x) \right| \, dx \le \frac{\epsilon}{6(M+2)}. \tag{70}$$

For any $f, g \in \mathcal{F}_1$ such that $\pi f = \pi g$, (67) tells us that $d(f, g)_{L^1}$ is at most $\epsilon/6(M+2)$. Recall $M$ was defined such that $d(\mathcal{R}[f], \mathcal{R}[g])_{L^1} \leq M d(f, g)_{L^1}$ for any $\mathcal{R}$.

$$d(\pi \circ \mathcal{R}[f], \pi \circ \mathcal{R}[g])_{L^1} \leq \frac{M\epsilon}{6(M+2)} + \frac{\epsilon}{6(M+2)} \tag{71}$$

$$= \frac{(M+1)}{(M+2)} \frac{\epsilon}{6} \tag{72}$$

So defining:

$$\tilde{\mathcal{R}} = \arg\min_{\mathscr{H}} d(\mathscr{H} \circ \pi[f], \pi \circ \mathcal{R}[f])_{\ell^1}, \tag{73}$$

we have

$$\left| \tilde{\mathcal{R}} \circ \pi[f] - \pi \circ \mathcal{R}[f] \right| \leq \frac{(M+1)}{(M+2)} \frac{\epsilon}{6}. \tag{74}$$

Then by (70),

$$\int_\Omega \left| \mathcal{R}[f](x) - \pi^{-1} \circ \tilde{\mathcal{R}} \circ \pi[f](x) \right| \, dx \leq \frac{\epsilon}{6(M+2)} + \frac{(M+1)}{(M+2)} \frac{\epsilon}{6} \tag{75}$$

$$= \frac{\epsilon}{6}. \tag{76}$$

$\square$

**Lemma B.6.** *Consider the extension of $\tilde{\mathcal{R}}$ to $\mathbb{R}^N \to \mathbb{R}^N$ in which each component of the output has the form:*

$$\tilde{\mathcal{R}}_j(\mathbf{f}) = \begin{cases} \mathcal{R}[\pi^{-1}\mathbf{f}](x) & \text{if } \mathbf{f} \in \pi \mathcal{F}_1 \\ 0 & \text{otherwise.} \end{cases} \tag{77}$$

*Then any finite measure $\nu$ on the measurable space $(\mathcal{F}_1, \mathscr{A})$ induces a finite measure $\mu$ on $(\mathbb{R}^N, \mathscr{L})$, and $\int_{\mathbb{R}^N} |\tilde{\mathcal{R}}_j(\mathbf{f})| \mu(d\mathbf{f}) < \infty$ for each $j$.*

*Proof.* Since the $\sigma$-algebra $\mathscr{A}$ on $\mathcal{F}_1$ is generated by $\pi$, the measure $\mu : \mu(\pi S) = \nu(S)$ for all $S \in \mathscr{A}$ is finite and defined w.r.t. the Lebesgue measurable sets on $\pi \mathcal{F}_1$. Since $\pi \mathcal{F}_1$ can be identified with a measurable subset of $\mathbb{R}^N$, $\mu$ can be naturally extended to $\mathbb{R}^N$. Doing so makes it absolutely continuous w.r.t. the Lebesgue measure on $\mathbb{R}^N$.

To show $\tilde{\mathcal{R}}_j(\mathbf{f})$ is integrable, it is sufficient to show it is bounded and compactly supported.

$\mathcal{F}_1$ is bounded in the $L^1$ norm. Thus by (67), $\pi \mathcal{F}_1$ is bounded in the normalized $\ell_1$ norm. The $\ell_1$ norm in $\mathbb{R}^N$ is strongly equivalent to the uniform norm, so there is some compact set $[-c, c]^N$, $c > 0$ for which the extension of $\pi \mathcal{F}_1$ to $\mathbb{R}^N$ vanishes, so $\text{supp}(\tilde{\mathcal{R}}_j(\mathbf{f})) \subseteq [-c, c]^N$.

Similarly, $\pi \mathcal{F}_1$ is bounded in the $\ell^1$ norm, hence there exists $c'$ such that $\tilde{\mathcal{R}}_j < c'$ for all $j$. $\square$

**Lemma B.7.** *For any finite measure $\mu$ absolutely continuous w.r.t. the Lebesgue measure on $\mathbb{R}^n$, $J \in L^1(\mu)$ and $\epsilon > 0$, there is a network $\mathcal{K}$ such that:*

$$\int_{\mathbb{R}^n} |J(\mathbf{f}) - \mathcal{K}(\mathbf{f})| \, \mu(d\mathbf{f}) < \frac{\epsilon}{2}. \tag{78}$$

*Proof.* The following construction is adapted from Lu et al. (2017). Since $J$ is integrable, there is a cube $E = [-c, c]^n$ such that:

$$\int_{\mathbb{R}^n \setminus E} |J(\mathbf{f})| \mu(d\mathbf{f}) < \frac{\epsilon}{8} \tag{79}$$

$$\|J - \mathbb{1}_E J\|_1 < \frac{\epsilon}{8}. \tag{80}$$

*Case 1: $J$ is non-negative on all of $\mathbb{R}^n$*

Define the set under the graph of $J|_E$:

$$G_{E,J} \triangleq \{(\mathbf{f}, y) : \mathbf{f} \in E, y \in [0, J(\mathbf{f})]\}. \tag{81}$$

$G_{E,J}$ is compact in $\mathbb{R}^{n+1}$, hence there is a finite cover of open rectangles $\{R'_i\}$ satisfying $\mu(\cup_i R'_i) - \mu(G_{E,J}) < \frac{\epsilon}{8}$ on $\mathbb{R}^n$. Take their closures, and extend the sides of all rectangles indefinitely. This results in a set of pairwise almost disjoint rectangles $\{R_i\}$. Taking only the rectangles $R = \{R_i : \mu(R_i \cap G_{E,J}) > 0\}$ results in a finite cover satisfying:

$$\sum_{i=1}^{|R|} \mu(R_i) - \mu(G_{E,J}) < \frac{\epsilon}{8}. \tag{82}$$

This implies:

$$\sum_{i=1}^{|R|} \mu(R_i) < \|J\|_1 + \frac{\epsilon}{8}, \tag{83}$$

and also,

$$\frac{\epsilon}{8} > \sum_{i=1}^{|R|} \int_{\mathbb{R}^n} \mathbb{1}_{R_i}(\mathbf{f}, J(\mathbf{f})) \, \mu(d\mathbf{f}) + \|J\|_1 \tag{84}$$

$$\geq \int_E |J(\mathbf{f}) - \sum_{i=1}^{|R|} \mathbb{1}_{R_i}(\mathbf{f}, J(\mathbf{f}))| \, \mu(d\mathbf{f}), \tag{85}$$

by the triangle inequality. For each $R_i = [a_{i1}, b_{i1}] \times \ldots [a_{in}, b_{in}] \times [\zeta_i, \zeta_i + y_i]$, let $X_i$ be its first $n$ components (i.e., the projection of $R_i$ onto $\mathbb{R}^n$). Then we have

$$\int_E |J(\mathbf{f}) - \sum_{i=1}^{|R|} y_i \mathbb{1}_{X_i}(\mathbf{f})| \, \mu(d\mathbf{f}) < \frac{\epsilon}{8}. \tag{86}$$

Let $Y(\mathbf{f}) \triangleq \sum_{i=1}^{|R|} y_i \mathbb{1}_{X_i}(\mathbf{f})$. By the triangle inequality,

$$\int_{\mathbb{R}^n} |J(\mathbf{f}) - \mathcal{K}(\mathbf{f})| \, \mu(d\mathbf{f}) \leq \|J - \mathbb{1}_E J\|_1 + \|\mathbb{1}_E J - Y\|_1 + \|\mathcal{K} - Y\|_1 \tag{87}$$

$$< \frac{\epsilon}{4} + \|\mathcal{K} - Y\|_1, \tag{88}$$

by (80) and (86). So it remains to construct $\mathcal{K}$ such that $\|\mathcal{K} - Y\|_1 < \frac{\epsilon}{4}$. Because $\mathbb{1}_{X_i}$ is discontinuous at the boundary of the rectangle $X_i$, it cannot be produced directly from a DI-Net (recall that all layers are continuous maps). However, we can approximate it arbitrarily well with a piece-wise linear function that rapidly ramps from 0 to 1 at the boundary.

For fixed rectangle $X_i$ and $\delta \in (0, 0.5)$, consider the inner rectangle $X_\delta \subset X_i$:

$$X_\delta = (a_1 + \delta(b_1 - a_1), b_1 - \delta(b_1 - a_1)) \times \cdots \times (a_n + \delta(b_n - a_n), b_n - \delta(b_n - a_n)), \tag{89}$$

where we omit subscript $j$ for clarity. Letting $b'_i = b_i - \delta(b_i - a_i)$, define the function:

$$T(\mathbf{f}) = \prod_{i=1}^{n} \frac{1}{\delta} \big[ \texttt{ReLU}(\delta - \texttt{ReLU}(\mathbf{f}_i - b'_i)) - \texttt{ReLU}(\delta - \texttt{ReLU}(\mathbf{f}_i - a_i)) \big], \tag{90}$$

where $\texttt{ReLU}(x) = \max(x, 0)$. $T(\mathbf{f})$ is a piece-wise linear function that ramps from 0 at the boundary of $X_i$ to 1 within $X_\delta$, and vanishes outside $X_i$. Note that

$$\|\mathbb{1}_X - T\|_1 < \mu(X) - \mu(X_\delta) \tag{91}$$

$$= (1 - (1 - 2\delta)^n)\mu(X), \tag{92}$$

if $\mu$ is the Lebesgue measure. $\delta$ may need to be smaller under other measures, but this adjustment is independent of the input $\mathbf{f}$ so it can be specified *a priori*.

Recall that the function we want to approximate is $Y(\mathbf{f}) = \sum_{i=1}^{|R|} y_i \mathbb{1}_{X_i}(\mathbf{f})$. We can build DI-Net layers $\mathcal{K} : \mathbf{f} \mapsto \mathcal{K}(\mathbf{f}) = \sum_{i=1}^{|R|} y_i T_i(\mathbf{f})$, since this only involves linear combinations and ReLUs. Then,

$$\|\mathcal{K} - Y\|_1 = \int_{\mathbb{R}^n} \sum_{i=1}^{|R|} y_i \left(T_i(\mathbf{f}) - \mathbb{1}_{X_i}(\mathbf{f})\right) d\mathbf{f} \tag{93}$$

$$= \sum_{i=1}^{|R|} y_i \|\mathbb{1}_{X_i} - T_i\|_1 \tag{94}$$

$$< (1 - (1 - 2\delta)^n) \sum_{i=1}^{|R|} y_i \mu(X_i) \tag{95}$$

$$= (1 - (1 - 2\delta)^n) \sum_{i=1}^{|R|} \mu(R_i) \tag{96}$$

$$< (1 - (1 - 2\delta)^n) \left(\|J\|_1 + \frac{\epsilon}{8}\right), \tag{97}$$

by (83). And so by choosing:

$$\delta = \frac{1}{2}\left(1 - \left(1 - \frac{\epsilon}{4}\left(\|J\|_1 + \frac{\epsilon}{8}\right)^{-1}\right)^{1/n}\right), \tag{98}$$

we have our desired bound $\|\mathcal{K} - Y\|_1 < \frac{\epsilon}{4}$ and thereby $\|J - \mathcal{K}\|_1 < \frac{\epsilon}{2}$.

*Case 2: J is negative on some region of $\mathbb{R}^n$*

Letting $J^+(\mathbf{f}) = \max(0, J(\mathbf{f}))$ and $J^-(\mathbf{f}) = \max(0, -J(\mathbf{f}))$, define:

$$G_{E,J}^+ \triangleq \{(\mathbf{f}, y) : \mathbf{f} \in E, y \in [0, J^+(\mathbf{f})]\} \tag{99}$$

$$G_{E,J}^- \triangleq \{(\mathbf{f}, y) : \mathbf{f} \in E, y \in [0, J^-(\mathbf{f})]\}. \tag{100}$$

As in (82), construct covers of rectangles $R^+$ over $G_{E,J}^+$ and $R^-$ over $G_{E,J}^-$ each with bound $\frac{\epsilon}{16}$ and $\mathbb{R}^n$ projections $X^+$, $X^-$. Let:

$$Y^+(\mathbf{f}) = \sum_{i=1}^{|R^+|} y_i^+ \mathbb{1}_{X_i^+}(\mathbf{f}) \tag{101}$$

$$Y^-(\mathbf{f}) = \sum_{i=1}^{|R^-|} y_i^- \mathbb{1}_{X_i^-}(\mathbf{f}) \tag{102}$$

$$Y = Y^+ - Y^- \tag{103}$$

We can derive an equivalent expression to (86):

$$\frac{\epsilon}{8} > \int_E \left|J(\mathbf{f}) - \sum_{i=1}^{|R^+|} y_i^+ \mathbb{1}_{X_i^+}(\mathbf{f}) + \sum_{i=1}^{|R^-|} y_i^- \mathbb{1}_{X_i^-}(\mathbf{f})\right| d\mathbf{f} \tag{104}$$

$$= \|\mathbb{1}_E J - Y\|_1. \tag{105}$$

Similarly to earlier, we use (80) and (105) to get:

$$\int_{\mathbb{R}^n} |J(\mathbf{f}) - \mathcal{K}(\mathbf{f})|\, d\mathbf{f} < \frac{\epsilon}{4} + \|\mathcal{K} - Y\|_1. \tag{106}$$

Choosing $T_i^+(\mathbf{f})$ and $T_i^-(\mathbf{f})$ the piece-wise linear functions associated with $X_i^+$ and $X_i^-$, and:

$$\mathcal{K}(\mathbf{f}) = \sum_{i=1}^{|R^+|} y_i^+ T_i^+(\mathbf{f}) - \sum_{i=1}^{|R^-|} y_i^- T_i^-(\mathbf{f}), \tag{107}$$

we have:

$$\|\mathcal{K} - Y\|_1 = \int_{\mathbb{R}^n} \left| \sum_{i=1}^{|R^+|} y_i^+ \left( T_i^+(\mathbf{f}) - \mathbb{1}_{X_i^+}(\mathbf{f}) \right) - \sum_{i=1}^{|R^-|} y_i^- \left( T_i^-(\mathbf{f}) - \mathbb{1}_{X_i^-}(\mathbf{f}) \right) \right| d\mathbf{f}, \tag{108}$$

applying the triangle inequality,

$$\leq \sum_{i=1}^{|R^+|} y_i^+ \left\| \mathbb{1}_{X_i^+} - T_i^+ \right\|_1 + \sum_{i=1}^{|R^-|} y_i^- \left\| \mathbb{1}_{X_i^-} - T_i^- \right\|_1 \tag{109}$$

$$< (1 - (1 - 2\delta^+)^n) \sum_{i=1}^{|R^+|} y_i^+ \mu(X_i^+) + (1 - (1 - 2\delta^-)^n) \sum_{i=1}^{|R^-|} y_i^- \mu(X_i^-) \tag{110}$$

$$< (1 - (1 - 2\delta^+)^n) \left( \|J^+\|_1 + \frac{\epsilon}{16} \right) + (1 - (1 - 2\delta^-)^n) \left( \|J^-\|_1 + \frac{\epsilon}{16} \right). \tag{111}$$

By choosing:

$$\delta^+ = \frac{1}{2} \left( 1 - \left( 1 - \frac{\epsilon}{8} \left( \|J^+\|_1 + \frac{\epsilon}{16} \right)^{-1} \right)^{1/n} \right) \tag{112}$$

$$\delta^- = \frac{1}{2} \left( 1 - \left( 1 - \frac{\epsilon}{8} \left( \|J^-\|_1 + \frac{\epsilon}{16} \right)^{-1} \right)^{1/n} \right), \tag{113}$$

and proceeding as before, we arrive at the same bounds $\|\mathcal{K} - Y\|_1 < \frac{\epsilon}{4}$ and $\|J - \mathcal{K}\|_1 < \frac{\epsilon}{2}$.

Putting it all together, Algorithm 1 implements the network logic for producing the function $\mathcal{K}$.

We can provide $x$ with access to $\mathbf{f}$ either through skip connections or by appending channels with the values $\{c + \mathbf{f}_k\}_{k=1}^n$ (which will be preserved under ReLU).

$\square$

**Theorem B.8** (Maps between functions). *For any Lipschitz continuous map $\mathcal{R} : \mathcal{F}_1 \to \mathcal{F}_1$, any $\epsilon > 0$, and any finite measure $\nu$ w.r.t. the measurable space $(\mathcal{F}_1, \mathscr{A}_{\epsilon,\mathcal{R}})$, there exists a DI-Net $\mathcal{T}$ that satisfies:*

$$\int_{\mathcal{F}_1} \|\mathcal{R}(f) - \mathcal{T}(f)\|_{L^1(\Omega)} \nu(df) < \epsilon. \tag{114}$$

*Proof.* If $\nu$ is not normalized, the discrepancy of our point set needs to be further divided by $\max\{\nu(\mathcal{F}_1), 1\}$. We assume for the remainder of this section that $\nu$ is normalized. Perform the construction of Lemma B.7 $N$ times, each with a tolerance of $\epsilon/2NK$, where $K$ is the Lipschitz constant of $\mathcal{R}$. Choose a partition of unity $\{\psi_j\}_{j=1}^N$ for which $\psi_j(x) = \mathbb{1}[x_k = \arg\min_{x' \in X} d(x, x')]$, and output $N$ channels with the values $\{\mathcal{K}_j(\mathbf{f})\psi_j(\cdot)\}_{j=1}^N$. By summing these channels we obtain a network $\tilde{\mathcal{K}}$ that fully specifies the desired behavior of $\tilde{\mathcal{R}} : \mathbb{R}^N \to \mathbb{R}^N$, with combined error:

$$\int_{\mathbb{R}^N} \left\| \tilde{\mathcal{R}}(\mathbf{f}) - \tilde{\mathcal{K}}(\mathbf{f}) \right\|_{\ell^1} \mu(d\mathbf{f}) < \frac{\epsilon}{2}. \tag{115}$$

Thus,

$$\int_{\mathcal{F}_1} \left| \frac{1}{|X|} \sum_{x' \in X} \tilde{\mathcal{R}} \circ \pi[f](x') - \tilde{\mathcal{K}} \circ \pi[f](x) \right| \nu(df) \le \frac{\epsilon}{2}. \tag{116}$$

By (68) we have:

$$\int_{\mathcal{F}_1} \left| \int_{\Omega} \left| \pi^{-1} \circ \tilde{\mathcal{R}} \circ \pi[f](x) - \pi^{-1} \circ \tilde{\mathcal{K}} \circ \pi[f](x) \right| dx \right| \nu(df) \le \frac{\epsilon}{2} + \frac{\epsilon}{6(M+2)} \tag{117}$$

By Lemma B.5 we have:

$$\int_{\mathcal{F}_1} \int_{\Omega} \left| \mathcal{R}[f](x) - \pi^{-1} \circ \tilde{\mathcal{K}} \circ \pi[f](x) \right| dx \, \nu(df) \le \frac{\epsilon}{2} + \frac{\epsilon}{6(M+2)} + \frac{\epsilon}{6} \tag{118}$$

And thus the network $\mathcal{T} = \pi^{-1} \circ \tilde{\mathcal{K}} \circ \pi$ gives us the desired bound:

$$\int_{\mathcal{F}_1} \|\mathcal{R}(f) - \mathcal{T}(f)\|_{L^1(\Omega)} \nu(df) < \epsilon. \tag{119}$$

$\square$

**Corollary B.9** (Maps from functions to vectors). *For any Lipschitz continuous map $\mathcal{R} : \mathcal{F}_1 \to \mathbb{R}^n$, any $\epsilon > 0$, and any finite measure $\nu$ w.r.t. the measurable space $(\mathcal{F}_1, \mathscr{A}_{\epsilon,\mathcal{R}})$, there exists a DI-Net $\mathcal{T}$ that satisfies:*

$$\int_{\mathcal{F}_1} \|\mathcal{R}(f) - \mathcal{T}(f)\|_{\ell_1(\mathbb{R}^n)} \nu(df) < \epsilon. \tag{120}$$

*Proof.* Let $M_0$ be the Lipschitz constant of $\mathcal{R}$ in the sense that $d(\mathcal{R}[f], \mathcal{R}[g])_{\ell^1} \le M_0 d(f, g)_{L^1}$. Let $M = \max\{M_0, 1\}$. There exists $\tilde{\mathcal{R}} : \pi\mathcal{F}_1 \to \mathbb{R}^n$ such that $\left\| \tilde{\mathcal{R}} \circ \pi[f] - \mathcal{R}[f] \right\|_{\ell^1} \le \epsilon/12$. As in Lemma B.6, consider the extension of $\tilde{\mathcal{R}}$ to $\mathbb{R}^N \to \mathbb{R}^n$ in which each component of the output has the form:

$$\tilde{\mathcal{R}}_j(\mathbf{f}) = \begin{cases} \mathcal{R}[\pi^{-1}\mathbf{f}]_j & \text{if } \mathbf{f} \in \pi\mathcal{F}_1 \\ 0 & \text{otherwise.} \end{cases} \tag{121}$$

Then for similar reasoning, $\nu$ on $\mathcal{F}_1$ induces a measure $\mu$ on $\mathbb{R}^N$ that is finite and absolutely continuous w.r.t. the Lebesgue measure, and $\int_{\mathbb{R}^N} |\tilde{\mathcal{R}}_j(\mathbf{f})| \mu(d\mathbf{f}) < \infty$ for each $j$.

We construct our $\mathbb{R}^N \to \mathbb{R}$ approximation $n$ times with a tolerance of $\epsilon/2n$, such that:

$$\int_{\mathbb{R}^N} \left\| \tilde{\mathcal{R}}(\mathbf{f}) - \tilde{\mathcal{K}}(\mathbf{f}) \right\|_{\ell^1(\mathbb{R}^n)} \mu(d\mathbf{f}) < \frac{\epsilon}{2}. \tag{122}$$

Applying (67), we find that the network $\mathcal{T} = \tilde{\mathcal{K}} \circ \pi$ gives us the desired bound:

$$\int_{\mathcal{F}_1} \|\mathcal{R}(f) - \mathcal{T}(f)\|_{\ell_1(\mathbb{R}^n)} \nu(df) < \epsilon. \tag{123}$$

$\square$

**Corollary B.10** (Maps from vectors to functions). *For any Lipschitz continuous map $\mathcal{R} : \mathbb{R}^n \to \mathcal{F}_1$ and any $\epsilon > 0$, there exists a DI-Net $\mathcal{T}$ that satisfies:*

$$\int_{\mathbb{R}^n} \|\mathcal{R}(x) - \mathcal{T}(x)\|_{L^1(\Omega)} dx < \epsilon. \tag{124}$$

*Proof.* Define the map $\tilde{\mathcal{R}} : \mathbb{R}^n \to \pi\mathcal{F}_1 \subset \mathbb{R}^N$ by $\tilde{\mathcal{R}} = \pi \circ \mathcal{R}$. Since $\tilde{\mathcal{R}}$ is bounded and compactly supported, $\int_{\mathbb{R}^N} |\tilde{\mathcal{R}}_i(x)|dx < \infty$ for each $i$.

We construct a $\mathbb{R}^n \to \mathbb{R}$ approximation $N$ times each with a tolerance of $\epsilon/2NK$ with $K$ the Lipschitz constant, such that:

$$\int_{\mathbb{R}^n} \left\| \tilde{\mathcal{R}}(x) - \tilde{\mathcal{K}}(x) \right\|_{L^1(\Omega)} dx < \frac{\epsilon}{2}. \tag{125}$$

Applying (68), we find that the network $\mathcal{T} = \pi^{-1} \circ \tilde{\mathcal{K}}$ gives us the desired bound:

$$\int_{\mathbb{R}^n} \left\| \mathcal{R}(x) - \mathcal{T}(x) \right\|_{L^1(\Omega)} dx < \epsilon. \tag{126}$$

$\square$

Consider the space of vector-valued functions $\mathcal{F}_c = \{f : \Omega \to \mathbb{R}^c : \int_\Omega \|f\|_1 dx < \infty, f_i \in \mathcal{F}_1 \text{ for each } i\}$. Denote the norm on this space as:

$$\|f\|_{\mathcal{F}_c} = \int_\Omega \sum_{i=1}^c |f_i(x)|dx. \tag{127}$$

**Definition B.11** (Concatenation)**.** Concatenation is a map from two scalar functions $f_i, f_j \in \mathcal{F}_1$ to the vector-valued function $[f_i, f_j] \in \mathcal{F}_2$. The concatenation of vector-valued functions can be defined inductively to yield $\mathcal{F}_n \times \mathcal{F}_m \to \mathcal{F}_{n+m}$ for any $n, m \in \mathbb{N}$.

All maps $\mathcal{F}_n \times \mathcal{F}_m \to \mathcal{F}_c$ can be expressed as a concatenation followed by a map $\mathcal{F}_{n+m} \to \mathcal{F}_c$. A map $\mathbb{R}^n \to \mathcal{F}_m$ is also equivalent to $m$ maps $\mathbb{R}^n \to \mathcal{F}_1$ followed by concatenation. Thus, we need only characterize the maps that take one vector-valued function as input.

Considering the maps $\mathcal{F}_n \to \mathcal{F}_m$, we choose a lower discrepancy point set $X$ on $\Omega$ such that the Koksma–Hlawka inequality yields a bound of $\epsilon/12mn(M + 2)$. Let $\pi$ project each component of the input to $\pi\mathcal{F}_1$, and $\pi^{-1}$ inverts this projection under some choice function. We take $\mathscr{A}'$ to be the product $\sigma$-algebra generated from this $\pi$: $\mathscr{A}' = \{E_1 \times \cdots \times E_c : E_1, \ldots, E_c \in \mathscr{A}\}$ where $\mathscr{A}$ is the $\sigma$-algebra on $\mathcal{F}_1$ from Definition B.4.

**Corollary B.12** (Maps between vector-valued functions)**.** *For any Lipschitz continuous map $\mathcal{R} : \mathcal{F}_n \to \mathcal{F}_m$, any $\epsilon > 0$, and any finite measure $\nu$ w.r.t. the measurable space $(\mathcal{F}_n, \mathscr{A}'_{\epsilon,\mathcal{R}})$, there exists a DI-Net $\mathcal{T}$ that satisfies:*

$$\int_{\mathcal{F}_n} \|\mathcal{R}(f) - \mathcal{T}(f)\|_{\mathcal{F}_m} \nu(df) < \epsilon. \tag{128}$$

*Proof.* The proof is very similar to that of Theorem B.8. Our network now requires $nN$ maps from $\mathbb{R}^{mN} \to \mathbb{R}$ each with error $\epsilon/2mnN$. Summing the errors across all input and output channels yields our desired bound. $\square$

The vector-valued analogue of Corollary B.9 is clear, and we state it here for completeness:

**Corollary B.13** (Maps from vector-valued functions to vectors)**.** *For any Lipschitz continuous map $\mathcal{R} : \mathcal{F}_n \to \mathbb{R}^m$, any $\epsilon > 0$, and any finite measure $\nu$ w.r.t. the measurable space $(\mathcal{F}_n, \mathscr{A}'_{\epsilon,\mathcal{R}})$, there exists a DI-Net $\mathcal{T}$ that satisfies:*

$$\int_{\mathcal{F}_n} \|\mathcal{R}(f) - \mathcal{T}(f)\|_{\ell_1(\mathbb{R}^m)} \nu(df) < \epsilon. \tag{129}$$

## C  Pixel-Based DI-Net Layers

Here we present a number of additional DI layers that show how to generalize pixel-based networks (convolutional neural networks and vision transformers) to DI-Net equivalents. Many of the following layers were not directly used in our experiments, and we leave an investigation of their properties for future work.

**Max pooling**  There are two natural generalizations of the max pooling layer to a collection of points: 1) assigning each point to the maximum of its k nearest neighbors, and 2) taking the maximum value within a fixed-size window around each point. However, both of these specifications change the output's behavior as the density of points increases. In the first case, nearest neighbors become closer together so pooling occurs over smaller regions where there is less total variation in the NF. In the second case, the empirical maximum increases monotonically as the NF is sampled more finely within each window. Because we may want to change the number of sampling points on the fly, both of these behaviors are detrimental.

If we consider the role of max pooling as a layer that shuttles gradients through a strong local activation, then it is sufficient to use a fixed-size window with some scaling factor that mitigates the impact of changing the number of sampling points. Consider the following simplistic model: assume each point in a given patch of an NF channel is an i.i.d. sample from $\mathcal{U}([-b, b])$. Then the maximum of $N$ samples $\{f_i(x_j)\}_{j=1}^N$ is on average $\frac{N-1}{N+1}b$. So we can achieve an "unbiased" max pooling layer by taking the maximum value observed in each window and scaling it by $\frac{N+1}{N-1}$ (if $N = 1$ or our empirical maximum is negative then we simply return the maximum), then (optionally) multiplying a constant to match the discrete layer.

To replicate the behavior of a discrete max pooling layer with even kernel size, we shift the window by half the dimensions of a pixel, just as in the case of convolution.

**Tokenization**  A tokenization layer chooses a finite set of non-overlapping regions $\omega_j \subset \Omega$ of equal measure such that $\cup_j \omega_j = \Omega$. We apply the indicator function of each set to each channel $f_i$. An embedding of each $f_i|_{\omega_j}$ into $\mathbb{R}^n$ can be obtained by taking its inner product with a polynomial function whose basis spans each $L^2(\omega_j)$. To replicate a pre-trained embedding matrix, we interpolate the weights with B-spline surfaces.

**Average pooling**  An average pooling layer performs a continuous convolution with a box filter, followed by downsampling. To reproduce a discrete average pooling with even kernels, the box filter is shifted, similarly to max pooling.

An adaptive average pooling layer can be replicated by tokenizing the NF and taking the mean of each token to produce a vector of the desired size.

**Attention layer**  There are various ways to replicate the functionality of an attention layer. Here we present an approach that preserves the domain. For some $d_k \in \mathbb{N}$ consider a self-attention layer with $c_{\mathrm{in}}d_k$ parametric functions $q_{ij} \in L^2(\Omega)$, $c_{\mathrm{in}}d_k$ parametric functions $k_{ij} \in L^2(\Omega)$, and a convolution with $d_k$ output channels, produce the output NF $g$ as:

$$Q_j = \langle q_{ij}, f_i \rangle \tag{130}$$

$$K_j = \langle k_{ij}, f_i \rangle \tag{131}$$

$$V[f] = \sum_{i=1}^{c_{\mathrm{in}}} v_{ij} * f_i + b_j \tag{132}$$

$$g(x) = \texttt{softmax}\left(\frac{QK^T}{\sqrt{d_k}}\right) V[f](x) \tag{133}$$

A cross-attention layer generates queries from a second input NF. A multihead-attention layer generates several sets of $(Q, K, V)$ triplets and takes the softmax of each set separately.

**Data augmentation**  Most data augmentation techniques, including spatial transformations, point-wise functions and normalizations, translate naturally to NFs. Furthermore, spatial transformations are efficient and do not incur the usual cost of interpolating back to the grid. Thus DI-Nets might be suitable for a new set of data augmentation methods such as adding Gaussian noise to the discretization coordinates.

**Positional encoding**  Given their central role in neural fields, positional encodings (adding sinusoidal functions of the coordinates to each channel) can help pixel-based DI-Nets learn high-frequency patterns under a range of discretizations.

**Vector decoders ($\mathbb{R}^n \to \mathcal{F}_c$) and parametric functions in $\mathcal{F}_c$**   A vector can be expanded into an NF in several ways. We can create an NF that simply places input values at fixed coordinates and produces values at all other coordinates by interpolation. Alternatively, we can define a parametric function that spans $\mathcal{F}_c$ using the input vector as the parameters, for example by taking as input $n$ numbers and treating them as coefficients of the first $n$ elements of an orthonormal polynomial basis on $\Omega$. If $\Omega$ is a subset of $[a, b]^d$, one can use a separable basis defined by the product of rescaled 1D Legendre polynomials along each dimension. If $\Omega$ is a $d$-ball, we can use the Zernike polynomial basis. For a general coordinate system, a small MLP could be used where $\mathbb{R}^n$ can represent its parameters or a learned lower-dimensional modulation (Dupont et al., 2022) of its parameters. Beyond using such parametric functions as vector decoder layers, they also give rise to $n$-parameter layers that compute an inner product ("learned global pooling layer") or elementwise product ("dense modulation layer") of an input NF with the learned functions.

**Warp layer**   Layers that apply a self-homeomorphism $q$ on $\Omega$ (a bicontinuous map from $\Omega \to \Omega$) preserve discretization invariance since it simply modifies the upper bound of the invariance error in subsequent layers to use a discrepancy of $q(X)$ rather than a discrepancy of $X$.

---

**Algorithm 2:** Classifier Training

**Input:** network $\mathcal{T}_\theta$, dataset $\mathcal{D}$, classifier loss $\mathcal{L}$, input discretization $X$
**for** step $s \in 1 : N_{\text{steps}}$ **do**
    Neural fields $f_i$, labels $y_i \leftarrow \text{minibatch}(\mathcal{D})$
    Label estimates $\hat{y}_i \leftarrow \mathcal{T}_\theta[f_i; X]$
    Update $\theta$ based on $\nabla_\theta \mathcal{L}(\hat{y}_i, y_i)$
**end**
**Output:** trained network $\mathcal{T}_\theta$

---

**Algorithm 3:** Dense Prediction Training

**Input:** network $\mathcal{T}_\theta$, dataset $\mathcal{D}$ with coordinate-label pairs, task-specific loss $\mathcal{L}$
**for** step $s \in 1 : N_{\text{steps}}$ **do**
    Neural fields $f_i$, point labels $(\boldsymbol{x}_{ij}, y_{ij}) \leftarrow \text{minibatch}(\mathcal{D})$
    Point label estimates $\hat{y}_{ij} \leftarrow \mathcal{T}_\theta[f_i; \{\boldsymbol{x}_{ij}\}_{j=1}^{N_j}](\boldsymbol{x}_{ij})$
    Update $\theta$ based on $\nabla_\theta \mathcal{L}(\hat{y}_{ij}, y_{ij})$
**end**
**Output:** trained network $\mathcal{T}_\theta$

---

