# OpenReview forum: "Discretization Invariant Networks for Learning Maps between Neural Fields"
_TMLR — Accepted by TMLR_

### Review · Reviewer_agB5 · 2023-08-01

**Summary Of Contributions:**

This paper proposes a simple methodology for learning a mapping function that takes a neural field as input and maps it to either another neural field (regression) of a discrete target (classification). The main benefit of the proposed methodology compared with existing techniques is that existing techniques either directly use the model, that encodes the neural field, as the input to the classification/regression model, or they use some discrete sampling of the neural field as the input. The former technique has the limitation that it cannot work across different encodings, while the latter does not generalize beyond the specific discretization used for discretizing the input.

Along with the proposed methodology, the authors also studies various aspects of discretization invariance of the proposed mapping function in the context of neural fields--

1. they define a formal notion of discretization invariance in terms of the variation of the underlying function and the discrepency of the discretization (which intuitively measures how far the discrete samples are from their continuous counterpart). Specificaly, absolute difference between the integral estimate and the sample estimate is upper bounded by the product of the variation function and the discrepancy function. This bound establishes the condition in which the discretization map is convergent, i.e., the absolute difference approaches zero, leading to discretiztion invariance. This condition requires the discretization to be equidistributed, i.e., the discrepancy function asymptotically tends to zero.

2. they also show that under such conditions, the gradients of the mapping function w.r.t. its parameters are also convergent, i.e., the gradients computed using the integral form and the discretized form are asymptotically identical.

3. they show that the proposed Discretization Invariant (DI) networks are universal approximators when the discretization method has a small enough discrepancy.

4. they discuss existing commonly used neural network layers that can be used as part of the proposed DI network.

5. they discuss the computational complexity of the proposed framework and claim that the complexity scales linearly with the number of points sampled in the discretization process.


Finally, the authors conduct experiments covering various aspects of the proposed methodology:
1. Classification performance of DI-Net across different resolutions.
2. Classification performance of DI-Net comparision when training and test time discretizations are different.
3. Segmentation performance comparision with other existing methods.
4. Regression performance of DI-Net comparision when training and test time discretizations are different.

**Audience:**

Yes

**Claims And Evidence:**

Yes

**Requested Changes:**

See weakness/concern above.

**Strengths And Weaknesses:**

Strengths:

1. As the authors point out, the key advantage of learning a DI network over neural fields as opposed to using a rendered image or using the neural field model parameter as inputs is that the framework becomes parameterization agnostic. This allows the model to work across different neural field parameterizations and discretizations. This is a useful property.

2. Experiments investigating several aspects of the proposed methodology and problem space.

3. Detailed analysis covering different aspects of the prediction problem when dealing with NFs.


Weakness/Concerns:

1. Since number of point sampled should grow quadratically with the image resolution, does this imply the complexity is quadratic w.r.t. the image resolution?

2. It would be helpful to discuss the complexity of the proposed method with methods that do not operate on the NF samples, but rather work either on the rendered image or the NF model parameters. This discussion is currently missing.

3. One comparison that I think is important but missing in the paper is to compare the performance of DI-Net with existing applicable methods when training and test time discretizations are different.

---

> ### Author Response · Authors · 2023-09-11
> **Response to Reviewer agB5**
>
> We thank the reviewer for their feedback. We are pleased that the reviewer found our paper had a number of strengths in the methodology, experiments, and analysis. We address their concerns as follows:
>
> *> "Since number of point sampled should grow quadratically with the image resolution, does this imply the complexity is quadratic w.r.t. the image resolution?"*
>
> Yes, for an implementation on a grid pattern, the complexity is O(*wh*) for width *w* and height *h*, which is the same scaling behavior as CNNs and Fourier neural operators. The term "image resolution" is not well-defined when the domain is not sampled in a grid pattern, and so DI-Net has the additional flexibility of being able to choose low discrepancy sequences or other sampling patterns which may scale better than a grid pattern. For example, PDE solvers for fluid motion can benefit from sampling more points close to object boundaries rather than on a uniform grid.
>
> *> "It would be helpful to discuss the complexity of the proposed method with methods that do not operate on the NF samples, but rather work either on the rendered image or the NF model parameters. This discussion is currently missing."*
>
> We revised our paper to provide a brief discussion of the efficiency of our method compared to learning directly on the rendered image or NF model parameters. This discussion is inserted at the end of section 5:
>
> "DI-Nets are similar in computational complexity to discrete neural networks, with the added flexibility of being able to sample in non-grid patterns that can converge more efficiently. Although networks that operate directly on the parameters of the neural field can access the entire NF in constant time, the NF itself needs more parameters to capture finer resolutions, and in practice many downstream tasks can be solved at a much coarser resolution than would be captured by the NF. Thus such networks do not necessarily scale better than DI-Nets, and would likely suffer from higher computational costs when there is a mismatch between the resolution of the NF and the resolution needed for the task."
>
> *> "One comparison that I think is important but missing in the paper is to compare the performance of DI-Net with existing applicable methods when training and test time discretizations are different."*
>
> Existing methods are fairly limited when it comes to accommodating arbitrary discretizations: most frameworks still require that the number and location of query points remain the same at training and test time; one of the exceptions, VIDON [5], can actually be considered a type of DI-Net. Indeed, one of the motivations of the DI-Net framework is to describe a class of networks with this unique capability. We included a comparison between DI-Nets and CNNs on classification when changing the grid resolution of the input image at test time (Fig. 3), although since DI-Net generally performs suboptimally on the grid compared to other sampling patterns (Tables 1 and 3), we believe this comparison is not of great practical importance.

---

### Review · Reviewer_13K2 · 2023-08-22

**Summary Of Contributions:**

This work studies the problem of discretization invariant learning for neural fields ( a kind of continuous data). Authors proposed a new framework for designing and understanding discretization invariant neural networks (DI-Nets) for learning maps between functions on continuous domains. The analysis establishes upper bounds on the deviation in model outputs under different finite discretizations and highlights the central role of point set discrepancy in characterizing such bounds. The authors design a family of neural networks driven by numerical integration via quasi-Monte Carlo sampling with discretizations of low discrepancy. The DI-Nets can universally approximate a large class of maps between integrable function spaces, and show discretization invariance in backpropagation through such models. The article demonstrates the effectiveness of DI-Nets in learning to classify and segment visual data under various discretizations, and sometimes generalize to new types of discretizations at test time.

**Audience:**

Yes

**Claims And Evidence:**

Yes

**Requested Changes:**

Please address my concerns in weaknesses.

**Strengths And Weaknesses:**

Strengths
1. The paper was well-motivated and well-written with good structures.
2. The proposal is a new framework for designing and understanding discretization invariant neural networks (DI-Nets) that can learn maps between functions on continuous domains without being sensitive to how the function is sampled. The authors establish upper bounds on the deviation in model outputs under different finite discretizations and highlight the central role of point set discrepancy in characterizing such bounds.
3. In contrast of theortical understandings, authors design a family of neural networks driven by numerical integration via quasi-Monte Carlo sampling with discretizations of low discrepancy, which universally approximate a large class of maps between integrable function spaces. The use of low discrepancy sampling can guide an initial discretization, and adaptive sampling techniques such as adaptive quadrature or sphere tracing can be used to refine the discretization
4. They also show that DI-Nets can be applied to neural fields to learn to classify and segment visual data under various discretizations and sometimes generalize to new types of discretizations at test time. The proposed soluton outperform baseline algorithms on neural field classification, neural field segmentation, signed distance function prediction tasks with certain margin.

Weaknesses
1. The proposed solution combines neural networks with numerical integration via quasi-Monte Carlo sampling with discretizations of low discrepancy. I am wondering if it is possible to use neural ODE-based solution or other NN-based solver to infer the numerical integration directly, rather than performing MCMC-based integration. In this, authors can form the solution as an end-to-end NN-based solution.
2. MCMC-based solution is considered to be inefficient, in terms of lattency or time consumption, especially when the number of dimensions is high. I am wondering if it is possible to report and analyze the time consumption for the field generation versus resolutions and spaces.

---

> ### Author Response · Authors · 2023-09-11
> **Response to Reviewer 13K2**
>
> We thank the reviewer for their feedback. We are pleased that the reviewer found our paper "well-motivated and well-written" with both theoretical and experimental strengths. We address their concerns as follows:
>
> *> "The proposed solution combines neural networks with numerical integration via quasi-Monte Carlo sampling with discretizations of low discrepancy. I am wondering if it is possible to use neural ODE-based solution or other NN-based solver to infer the numerical integration directly, rather than performing MCMC-based integration. In this, authors can form the solution as an end-to-end NN-based solution."*
>
> This is an interesting line of thought. Although neural ODEs are not suitable for this setting (they integrate over a single dimension, and are designed to learn unknown latent dynamics rather than to estimate a known integral), there may be some other ways to speed up or amortize the integral computation. Performing simple Monte Carlo integration is typically quite fast – MCMC is not part of its computation. However, it may be possible to reduce the number of samples needed over the course of training or at inference time, and we revise our paper to discuss such extensions in Section 7 (Future Directions):
>
> "It may also be possible to reduce the number of samples needed to achieve low discretization error over the course of training or at inference time, for example by learning which discretizations produce more reliable estimates, or by designing layers that encourage the integral to converge to a predefined set of quantized values and propagating these quantized values downstream."
>
> *> "MCMC-based solution is considered to be inefficient, in terms of lattency or time consumption, especially when the number of dimensions is high. I am wondering if it is possible to report and analyze the time consumption for the field generation versus resolutions and spaces."*
>
> We clarify that our solution does not involve MCMC, which is distinct from (quasi) Monte Carlo integration. Our approach indeed does not scale well to discretizations of high dimensional spaces, but almost all applications of neural fields are low-dimensional, for example NeRF is 5-dimensional. We also note that existing architectures (e.g. CNNs, Fourier neural operators) are not designed to operate on data with more than 3 or 4 dimensions. At the bottom of section 5.3 (Convolutional DI-Nets), we mention that time complexity of DI-Nets scales linearly in the numbers of points sampled. The number of points / resolution needed for a given space may vary significantly depending on the task. For example, many 3D scene understanding tasks may be solved by projecting an irregular point cloud onto 2D planes corresponding to a set of cameras. Similar techniques can be used to perform fast integration over lower dimensional projections of the original function space. We have expanded the “Computational Complexity” paragraph at the end of section 5 into a subsection, with a more thorough discussion about the time complexity of our method compared to alternative methods.

---

### Review · Reviewer_Pe3A · 2023-08-28

**Summary Of Contributions:**

This paper presents DI-Nets, a framework for building in "discretization-invariance" into neural fields. Such models are particularly relevant in a bunch of different application settings, including high-fidelity computer graphics. The key technical contributions are:
- a universal approximation-style result for a large family of discretization-invariant networks
- analysis of backprop for such networks
- experimental demonstration of their capabilities in classification and segmentation
- experimental demonstration of "invariance" .

**Audience:**

No

**Claims And Evidence:**

Yes

**Requested Changes:**

**(Critical)**

- Please engage with the (theoretical) literature on continuous operator learning better (see Bartolucci et al above and many references therein). Motivate why your new framework is needed; e.g. how the point set discrepancy idea solves some of the issues in earlier work, practical aspects of implementing this idea and how you resolved them, etc.
- Consider including at least one motivating application where discretization invariance plays a key role; I don't believe segmentation on Cityscapes fits this setting tbh.

**(Other)**

- Consider highlighting limitations and barriers to "true" discretization equivalence, and possible ways forward.

**Strengths And Weaknesses:**

**Strengths**

- The direction is important and timely, particularly with the prominent rise of neural field models in many applications.
- The paper includes a thorough and precise mathematical treatment.

**Weaknesses**

- While these are not supposed to be criteria for TMLR, I'll mention this anyway: I am not very positive about this paper in terms of either novelty or impact.
    * In terms of novelty, there are similar treatments in the (recent) literature on understanding and trying to mitigate out the effects of discretization, particularly in the context of operator learning. Why is a new framework needed at all?
      The authors state that (p1) _" a general theory of discretization invariant deep learning has not yet been explored ..."_ and point to Kovachki et al. (2021) as an exception, but this paints a woefully incomplete picture since there is a wide body of literature on this topic. See, for example, Bartolucci et al., "Are Neural Operators Really Neural Operators?" (and also the numerous other papers referred to in this work.) The representation-equivalent neural operator (ReNO) framework should serve as a starting point.
    * In terms of impact, the main contributions are nice but somewhat uninteresting. (This is perhaps a byproduct of being too general a framework.)
          - Universal approximation: fine, but not surprising; the proof of Theorem 4.1 is AFAIK the usual rectangle-covering idea employed in UAT-style proofs. If the authors truly believe FNOs or attention layers can guide further design of DI-Nets they should investigate this further in detail.
          - Implementation: again, fine but didn't see too many new insights. Algorithms 2 and 3 are utterly uninformative and I would either remove them or restate in a more interesting way.
           - Experiments: all shown on super-shallow networks. Image classification and segmentation are anyway such mature problems that the techniques here are probably never going to be used. I'd recommend focusing on applications where discretization effects play a bigger role, and invariance to discretization is a key criterion (perhaps turbulent fluid flows? I am not sure).

Therefore, I give a "no" answer to the "Audience" question below, but am willing to debate this with the authors.
- I would have also liked to see a more honest discussion on the framework's limitations. It is impossible to achieve true, universal discretization-invariance in practice, since at some point effects with quantization and spatio-temporal resolution kick in. Where does the framework break down, and what is a likely path forward in such cases?

---

> ### Author Response · Authors · 2023-09-11
> **Response to Reviewer Pe3A (1/3)**
>
> We thank the reviewer for their feedback. We are pleased that they find our paper presents a "thorough and precise mathematical treatment", which is indeed the most important criteria for acceptance to TMLR. We have revised the introduction and related work sections based on the reviewer’s feedback in order to better clarify our novelty and impact. We respond to individual comments below:
>
> *> "In terms of novelty, there are similar treatments in the (recent) literature on understanding and trying to mitigate out the effects of discretization, particularly in the context of operator learning. Why is a new framework needed at all? The authors state that (p1) " a general theory of discretization invariant deep learning has not yet been explored ..." and point to Kovachki et al. (2021) as an exception, but this paints a woefully incomplete picture since there is a wide body of literature on this topic. See, for example, Bartolucci et al., "Are Neural Operators Really Neural Operators?" (and also the numerous other papers referred to in this work.) The representation-equivalent neural operator (ReNO) framework should serve as a starting point."*
>
> We believe our treatment of discretization invariance is unique and novel, as existing frameworks simply do not account for the choice of discretization when characterizing the approximation error of the learned operator. Our focus on the role of point set discrepancy fills an important gap in the literature. We expand the related work section to clarify this point:
>
> "[...] operator learning frameworks typically quantify the approximation error in terms of the class of functions being approximated or the type of layers used in the network (Kovachki et al., 2021b; Bartolucci et al., 2023; Kissas et al., 2022; Prasthofer et al., 2022) rather than the choice of discretization, a gap that we seek to fill in this work."
>
> "The recent operator learning framework ReNO (Bartolucci et al., 2023) starts with the assumption that there exists a lossless discretization of the input and output function spaces which is known a priori (e.g., they are bandlimited functions), then establishes necessary condition for learning (lossless) operators between such spaces. Other works are concerned with more practical aspects of operator learning: LOCA (Kissas et al., 2022) leverages attention to more efficiently learn correlations between related points in output space; NOMAD (Seidman et al., 2022) aims to increase expressivity given finite basis elements by equipping neural operators with nonlinear decoders; VIDON (Prasthofer et al., 2022) builds on DeepONet to accommodate arbitrary locations and numbers of input and output query points (as does our framework); other extensions of DeepONet and neural operators refine the original works to improve learning efficiency or generalizability on certain domains (Li et al., 2020b; Lu et al., 2021b; Lee et al., 2022)."
>
> In particular, the restrictions of ReNOs on the class of functions and choice of discretizations are too limited for many applications of interest. In contrast, DI-Nets apply to a much wider class of input/output function spaces and permit any choice of input/output discretization. We offer a new yet intuitive observation – the approximation error of the operator will be high when the input and/or output discretizations are sparse.
>
> Additionally, our perspective on discretization in invariant learning is introduced in the new context of learning on neural fields, where the effect of different discretizations is particularly relevant. We revise the introduction to clarify this point:
>
> "Current analyses of discretization invariance are limited to showing the existence of operators that converge in the limit of discretizations with infinite points [1], or they demand that the function spaces be constrained to those that can be discretized losslessly [2]. They do not examine the effect of the discretization itself on the approximation error in the general case. Characterizing this effect is particularly relevant in the context of neural fields, which permit many different types of discretizations and are often queried repeatedly under slightly different discretizations in applications such as novel view synthesis."

---

> > ### Author Response · Authors · 2023-09-11
> > **Response to Reviewer Pe3A (2/3)**
> >
> > *> "While these are not supposed to be criteria for TMLR, I'll mention this anyway: I am not very positive about this paper in terms of either novelty or impact. … Therefore, I give a "no" answer to the "Audience" question below, but am willing to debate this with the authors."*
> >
> > We argue that our paper fully meets the bar for audience interest. As discussed above, ours is the first work to draw a connection between the specific choice of discretization and the approximation error of the learned operator. We believe the lack of previous work in this direction indicates that our paper offers a fresh and informative perspective for many readers. We also argue that drawing a connection between operator learning and learning on neural fields is interesting for researchers on both topics, as to our knowledge this connection has not been made in previous work.
> >
> > *> "Universal approximation: fine, but not surprising; the proof of Theorem 4.1 is AFAIK the usual rectangle-covering idea employed in UAT-style proofs."*
> >
> > Rectangle covering is the most involved step in the proof of Theorem 4.1, although equally important in our context are steps 1 and 5 where the approximation error is influenced by the discrepancy of the chosen discretization, which is not accounted for in most UAT-style proofs in the discretization invariant learning literature.
> >
> > *> "Implementation: again, fine but didn't see too many new insights. Algorithms 2 and 3 are utterly uninformative and I would either remove them or restate in a more interesting way."*
> >
> > The implementation section is intended to make DI-Nets more concrete, and to clarify for the reader exactly how such networks are trained and used throughout our experiments, rather than to produce insight per se. In our revisions we have moved Algorithms 2 and 3 to the appendix. They were originally included to illustrate the similarity of DI-Net training to the discrete case.
> >
> > *> "I would have also liked to see a more honest discussion on the framework's limitations. It is impossible to achieve true, universal discretization-invariance in practice, since at some point effects with quantization and spatio-temporal resolution kick in. Where does the framework break down, and what is a likely path forward in such cases? … Consider highlighting limitations and barriers to "true" discretization equivalence, and possible ways forward."*
> >
> > The extent to which discretization invariance breaks down in practice is tested and discussed in several experiments, including Figure 3 (the effect of test time resolution on classification accuracy), Table 1 (the effect of different train/test discretizations on classification accuracy), and Table 3 (the effect of different train/test discretizations on SDF prediction). We also report and discuss several other limitations and negative findings of our framework in Sections 6.4 (Initialization with discrete networks) and 7 (Future Directions). To bridge the empirical gap in discretization invariance, we believe that our first point in Future Directions, "discretization invariance as learning signal", is the most promising approach. A DI-Net could be trained in an unsupervised manner to map different discretizations of each datapoint to similar features or outputs. Such a data-driven approach could greatly improve the approximation abilities of the network beyond theoretical guarantees. Limitations in spatio-temporal resolution and quantization relate to the point spread function of the sensors used to collect input and output data. In applications where this is the limiting factor on discretization invariance, a task-specific solution is likely required. We discuss this case in the last point in Future Directions, "error propagation".

---

> > > ### Author Response · Authors · 2023-09-11
> > > **Response to Reviewer Pe3A (3/3)**
> > >
> > > *> "- Experiments: all shown on super-shallow networks. Image classification and segmentation are anyway such mature problems that the techniques here are probably never going to be used. I'd recommend focusing on applications where discretization effects play a bigger role, and invariance to discretization is a key criterion (perhaps turbulent fluid flows? I am not sure). … Consider including at least one motivating application where discretization invariance plays a key role; I don't believe segmentation on Cityscapes fits this setting tbh."*
> > >
> > > We choose image classification and segmentation as toy examples of neural field applications, where the 2D setting makes it easier to examine the properties of such networks. Using simple shallow networks allows us to compare discretization effects between DI-Nets and baselines with matching architectures without needing to introduce regularizers and other optimizations to make DI-Nets scalable. We note that many other theoretical works introducing new techniques for manipulating neural fields also focus on simple experiments with 2D images [6, 7], even when the most relevant applications are 3D scenes (NeRF). Discretization plays a major role in processing neural fields such as NeRFs, as they are designed to produce faithful signals from a wide range of discretizations. However, further investigation is needed to scale DI-Nets to work reliably in the setting of 3D scenes, as deep DI-Nets are difficult to optimize (see Section 6.4). We revise the introduction, future directions and conclusion to describe more motivating applications where discretization invariant learning could play a key role. For example, we write:
> > >
> > > "Discretization invariance may be a particularly useful concept to harness in the context of 3D scene understanding, as the information in a scene that is relevant for most tasks of interest is invariant under a much wider range of discretizations (e.g., points, rays and light fields; 360 degree or forward-facing) than can be described purely by group symmetries."

---

### Author Response · Authors · 2023-09-11
**General response to reviewers**

We thank the reviewers for their thoughtful and detailed feedback which has helped to improve our submission. We are happy that all reviewers agreed that our work is supported by accurate, convincing and clear evidence, and a majority also agree that the paper is of interest to TMLR’s audience. We revised our paper to provide further motivation for our work and analysis of our method, and updated text is marked in blue.

---

### Decision · Action_Editors · 2023-10-13

**Recommendation:** Accept as is

**Comment:**

This paper examines discretization-invariant networks for learning maps between neural fields. Among other results, it proves upper bounds on how much model outputs can deviate under different finite discretizations and establishes universal approximation for a large class of maps between integrable function spaces. Empirical results are presented which demonstrate enhanced discretization-invariance relative to standard CNN models.

The reviewers generally agree that the topic is salient and that the mathematical results are sound. One weakness highlighted in the discussion concerned the novelty/impact. These concerns were effectively addressed by the authors in their response, and, moreover, as the reviewer noted, are not primary criteria for evaluation in any case. The claims of this paper are well supported by mathematical and empirical evidence, and on those grounds I support acceptance.

**Audience:**

Yes

**Claims And Evidence:**

Yes